# Calibration Attacks: A Comprehensive Study of Adversarial Attacks on Model Confidence

**Stephen Obadinma**                                                    *16sco@queensu.ca*
*Department of Electrical and Computer Engineering & Ingenuity Labs Research Institute, Queen's University*

**Xiaodan Zhu**                                                    *xiaodan.zhu@queensu.ca*
*Department of Electrical and Computer Engineering & Ingenuity Labs Research Institute, Queen's University*

**Hongyu Guo**                                                    *hongyu.guo@uottawa.ca*
*Digital Technologies Research Centre, National Research Council Canada*

**Reviewed on OpenReview:** *https://openreview.net/forum?id=TXzz9xwdpv*

## Abstract

In this work, we highlight and perform a comprehensive study on *calibration attacks*, a form of adversarial attacks that aim to trap victim models to be heavily miscalibrated without altering their predicted labels, hence endangering the trustworthiness of the models and follow-up decision making based on their confidence. We propose four typical forms of calibration attacks: *underconfidence*, *overconfidence*, *maximum miscalibration*, and *random confidence attacks*, conducted in both black-box and white-box setups. We demonstrate that the attacks are highly effective on both convolutional and attention-based models: with a small number of queries, they seriously skew confidence without changing the predictive performance. Given the potential danger, we further investigate the effectiveness of a wide range of adversarial defence and recalibration methods, including our proposed defences specifically designed for calibration attacks to mitigate the harm. From the ECE and KS scores, we observe that there are still significant limitations in handling calibration attacks. To the best of our knowledge, this is the first dedicated study that provides a comprehensive investigation on calibration-focused attacks. We hope this study helps attract more attention to these types of attacks and hence hamper their potential serious damages. To this end, this work also provides detailed analyses to understand the characteristics of the attacks. [1]

## 1 Introduction

While recent machine learning models have significantly improved the state-of-the-art performance on a wide range of tasks (Bengio et al., 2021; Vaswani et al., 2017; LeCun et al., 2015; Krizhevsky et al., 2012), these models are often vulnerable and easily deceived by perturbed inputs (Ren et al., 2020). Adversarial attacks (Ren et al., 2020) have been shown to be a crucial tool to reveal the susceptibility of victim models (Ibitoye et al., 2019; Zimmermann et al., 2022; Xiao et al., 2023). In the classic setup, adversarial examples are generated by introducing imperceptible modifications to an original datapoint to cause *misclassification*, where the focus is on trapping victim models to make incorrect predictions.

In this paper, we highlight and provide a comprehensive study on a different type of threat, which we call *calibration attacks*. These attacks focus on manipulating the victim models' confidence scores without modifying their predicted labels, hence endangering any follow-up decision-making that is based on the victim

---

[1]Our code is available at https://github.com/PhenetOs/CalibrationAttack

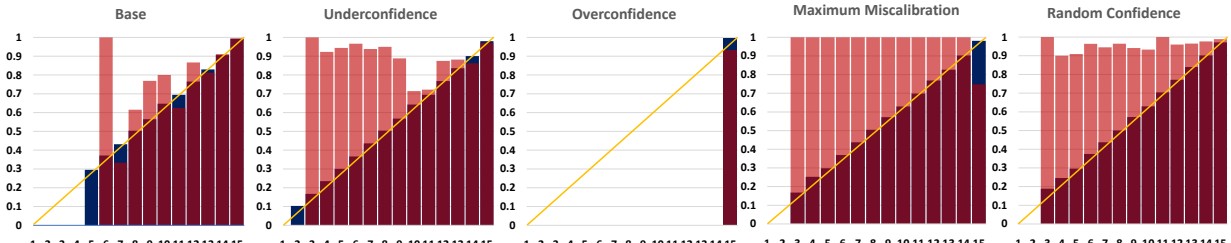

Figure 1: Reliability diagrams of a ResNet-50 classifier (fine-tuned and tested on Caltech-101) before and after the four forms of calibration attacks. Red bars show the average accuracy on the test data binned by confidence scores (15 bins) and the blue bars are the average confidence of samples in each bin. The x-axis represents the bins and y-axis is the accuracy (for red bars) or confidence (for blue bars). The yellow line represents perfect calibration. To have the minimum possible ECE the red bars and blue bars have to completely overlap in each bin (shown in maroon), where no overlap represents miscalibration. Despite the accuracy being unchanged, the miscalibration is severe after the attacks.

models' confidence. We propose to conduct four forms of calibration attacks: *underconfidence*, *overconfidence*, *maximum miscalibration*, and *random confidence attacks*, which can seriously skew confidence and cause heavy miscalibration, as demonstrated in the reliability diagrams in Figure 1. As we will show and discuss in our study, calibration attacks are insidious and hard to detect. The intrinsic harm is that on the surface the models appear to still make correct decisions, but the level of miscalibration could make the models' decisions malicious for downstream tasks.

Our specific studies consist of four forms of attacks, spanning black-box and white-box setups. We attack typical convolutional and attention-based models, and investigate both attack and defence methods.

In summary, our main contributions are as follows.

- To the best of our knowledge, this is the first dedicated study that provides a comprehensive investigation on confidence-focused calibration attacks.

- We propose four typical forms of calibration attacks and demonstrate their effectiveness and danger to victim models from different perspectives. Detailed insights are provided to understand the characteristics of the attacks and the vulnerability of victim models.

- We further investigate the effectiveness of a wide range of adversarial defence and recalibration methods, including our proposed defences specifically designed for calibration attacks to mitigate the harm. We hope our work helps attract more attention to these attacks and hence hamper their potential serious damages in applications.

## 2 Related Work

**Calibration of Machine Learning Models.** Among the two typical types of calibration methods, post-calibration methods are applied directly to the predictions of fully trained models at test time, which include classical approaches such as temperature scaling (Guo et al., 2017), Platt scaling (Platt, 1999), isotonic regression (Zadrozny & Elkan, 2002), and histogram binning (Zadrozny & Elkan, 2001). Training-based approaches, however, often add bias to help calibrate a model during training (Zhang et al., 2018; Thulasidasan et al., 2019; Kumar et al., 2018; Tomani & Buettner, 2021). We investigate a diverse range of calibration methods against calibration attacks to reveal the limitations that require them to be overhauled to deal with attacks, examining the vulnerability of models similar to those on the convolutional architectures (Guo et al., 2017; Minderer et al., 2021) and Transformer-based frameworks (Dosovitskiy et al., 2021). (Refer to Appendix A for a more detailed summary of related work.)

**Adversarial Attacks and Training.** Adversarial attacks are divided into black-box and white-box approaches. The former (Carlini & Wagner, 2017) assume less information about victim models. Many

methods are based on gradient estimation through querying the models and finding the finite differences (Bhagoji et al., 2018). White-box attacks often have access to the full details of a victim model such as its architecture and gradients (Goodfellow et al., 2015). Adversarial training and defence approaches have been introduced to improve the robustness of victim models (Chen et al., 2022; Qin et al., 2021; Patel et al., 2021; Dhillon et al., 2018). Related to these are works on generating certified robustness guarantees Kumar et al. (2020), where certified radii are generated for the predicted confidence of a smoothed classifier. The research in Emde et al. (2023) further extends this by certifying calibration through generating the worst-case bounds of calibration error, and discuss the importance studying attacks targeting calibration. Stutz et al. (2020) develop a modified adversarial training objective which focuses on biasing adversarial samples to be low-confidence predictions that can be easily filtered. Although this causes adversarial examples to be low confidence, it does not specifically target making low or high confidence adversarial examples through perturbations as a form of attack. In this work, we propose two defence models against calibration attacks.

**Attacking Uncertainty Estimates.** Galil & El-Yaniv (2021) first identified attacks on credible uncertainty estimates as an issue. Their models harm the potential of using uncertainty estimation on a model's predictions by pushing correct datapoints closer to the decision boundary and incorrect ones further away using confidence without causing incorrect predictions. Zeng et al. (2023) introduce a data poisoning attack designed to alter the training process of models such that a high-confidence region forms around an out-of-domain datapoint, harming the adversary-resistant uncertainty estimates on a set of targeted out-of-domain datapoints while leaving the original labels. Nevertheless, none of the prior works provide a comprehensive study on the calibration attacks, nor do they systematically investigate how well victim models would remain calibrated under such attacks, including under a range of different defence methods. Our work proposes and examines the effects of different attacks, the difficulty of detecting them, and the effectiveness of defence and calibration methods. We provide the study on both convolutional and more recent attention-based models, under both black and white-box approaches. Detailed analyses and insights are additionally discussed.

## 3 Calibration Attacks

Given the input $\mathbf{X} \equiv \{\mathbf{x}_1, \ldots, \mathbf{x}_N\}$ of $N$ datapoints and their ground-truth labels $\boldsymbol{\mathcal{Y}} \equiv \{y_1, \ldots, y_N\}$, where $y_i \in \{1, \ldots, K\}$ with $K$ being the number of classes, a classifier $\boldsymbol{\mathcal{F}}$ makes prediction for an instance $\mathbf{x}_i \in \mathbf{X}$ through a mapping $\boldsymbol{\mathcal{F}} : \mathbf{x}_i \to \langle \hat{y}_i, \hat{\boldsymbol{p}}_i \rangle$, where $\hat{y}_i$ is the predicted label which is often obtained by taking *argmax* on the output distribution $\hat{\boldsymbol{p}}_i$ over the $K$ classes: $\hat{y}_i = \mathrm{argmax}_{j=1}^K (\hat{p}_{ij})$. When needed, $\hat{\boldsymbol{p}}_i$ is written as $\hat{\boldsymbol{p}}(\mathbf{x}_i)$ and $\hat{\boldsymbol{p}}(\tilde{\mathbf{x}}_i)$, for the input $\mathbf{x}_i$ and its perturbation $\tilde{\mathbf{x}}_i$, respectively. Similarly, $\hat{y}_i$ can be rewritten as $\hat{y}(\mathbf{x}_i)$ or $\hat{y}(\tilde{\mathbf{x}}_i)$, which will be constrained to be same in calibration attacks.

### 3.1 Objective of Calibration Attacks

Calibration attacks aim to generate adversarial examples to optimize a predefined *miscalibration* function $\mathcal{M}(\tilde{\mathbf{x}}_i, k)$ for an adversarial example $\tilde{\mathbf{x}}_i$ and the predicted class $k$. As will be detailed below, we propose four forms of calibration attacks where $\mathcal{M}(\tilde{\mathbf{x}}_i, k)$ takes different implementations. Following conventional notation, an adversarial example $\tilde{\mathbf{x}}_i$ is created by adding noise $\boldsymbol{\delta}$ to an input $\mathbf{x}_i$: $\tilde{\mathbf{x}}_i = \mathbf{x}_i + \boldsymbol{\delta}$, bounded by $\epsilon$ in a $l_m$-ball:

$$\|\tilde{\mathbf{x}}_i - \mathbf{x}_i\|_m < \epsilon, \ m \in \{0, 1, \ldots, \infty\}, \tag{1}$$

where $\epsilon$ controls the amount of allowed perturbation and $m$ corresponds to different norms that may be used. In general, our attacks are based on the most popular view of class-wise calibration (Guo et al., 2017; Kull et al., 2019). For a datapoint $\langle \mathbf{x}_i, y_i \rangle$ in the dataset $\mathcal{D} = \{\langle \mathbf{x}_n, y_n \rangle\}_{n=1}^N$, a well calibrated model aims to achieve:

$$\mathbb{P}(y_i = k \mid \hat{p}_k(\mathbf{x}_i) = q_k) = q_k, \tag{2}$$

where $q_k$ is the confidence of the predicted class $k$ for $\mathbf{x}_i$. Any mismatch between the left and right hand sides of the equation creates undesirable miscalibration.

### 3.2 Four Forms of Calibration Attacks

Building on the above notations and framework, we propose four approaches to cover the most typical variants of calibration attacks.

**Underconfidence and Overconfidence Attacks (UCA and OCA).** These two types of attacks aim to solve the constrained optimization problem involving the miscalibration function $\mathcal{M}(\tilde{\mathbf{x}}_i, k)$, making a victim model either underconfident or overconfident.

$$\mathcal{M}_{UCA}(\tilde{\mathbf{x}}_i, k) = \hat{p}_k(\tilde{\mathbf{x}}_i) - \max_{j \neq k} \hat{p}_j(\tilde{\mathbf{x}}_i), \tag{3}$$

$$\mathcal{M}_{OCA}(\tilde{\mathbf{x}}_i, k) = 1 - \hat{p}_k(\tilde{\mathbf{x}}_i), \tag{4}$$

$$s.t. \ \hat{y}(\tilde{\mathbf{x}}_i) = \hat{y}(\mathbf{x}_i). \tag{5}$$

Calibration attacks focus on attacking the confidence of victim models without altering the predicted labels, which is satisfied by the constraint in Eq. 5.

By minimizing the loss $\mathcal{M}_{UCA}$ or $\mathcal{M}_{OCA}$, the attack models maximize calibration errors in these two setups respectively, with the corresponding adversarial examples $\tilde{\mathbf{X}}$ :

$$\max_{\tilde{\mathbf{X}}}(\mathbb{P}(y_i = k \mid \hat{p}_k(\tilde{\mathbf{x}}_i) = q_k) - q_k). \tag{6}$$

Algorithm 1 depicts an overview of calibration attacks, including UCA and OCA, but also MMA and RCA that we will introduce below. Unlike conventional attacks that modify the predicted labels and focus mainly on correctly classified examples, calibration attacks modify confidence and focus on both the originally correctly classified and misclassified instances. As detailed later in Section 4, we will implement calibration attacks in popular black-box (Andriushchenko et al., 2020) and white-box frameworks (Madry et al., 2018).

**Maximum Miscalibration Attacks (MMA).** We propose MMA with the aim of exploring and understanding more serious scenarios of miscalibration. The main principle of MMA is to perturb a set of inputs such that *(i)* all incorrectly classified datapoints (a set where the model has zero accuracy) have 100% confidence, and *(ii)* all correctly classified datapoints (a set where the model is 100% accurate) have the minimum possible confidence. MMA uses a combination of OCA and UCA to accomplish this, as shown in Algorithm 1. Proposition 3.1 below states the property of MMA in terms of the oracle (upper-bound) ECE score that can be achieved in theory, with the proof provided in Appendix C.

**Proposition 3.1.** *Assume $q$ is the accuracy of a K-way classifier $\mathcal{F}$ on the dataset $\mathcal{D} = \{\langle \mathbf{x}_n, y_n \rangle\}_{n=1}^N$. The Maximum Miscalibration Attack (MMA) maximizes the expected calibration error (ECE). The upper bound of ECE that can be achieved by MMA is $1 - q/K$.*

**Random Confidence Attacks (RCA).** We propose to perform RCA to decouple the victim model's confidence scores and predictive labels, in which the confidence scores produced by the attack model are randomized. RCA is performed by choosing a random target confidence score for each input, and then, depending on the original confidence score, running the corresponding underconfidence or overconfidence attacks to produce the target confidence score. Although RCA is theoretically less effective than MMA, it is less predictable, because unlike the other three types of attacks, RCA does not produce a predetermined direction (i.e., under- or overconfidence) of attacks for a given input. The produced confidence scores are often less extreme and more reasonable-looking, but largely meaningless due to randomization. Note that in addition to MMA and RCA, one may design other approaches to combine UCA with OCA, but MMA and RCA represent the two most typical compositions that create very harmful patterns of miscalibration.

### 3.3 Defence Against Calibration Attacks

As we will show in the experiments, calibration attacks are very effective. In addition to studying a wide range of existing defence methods, we introduce new methods specifically for calibration attacks: *Calibration Attack Adversarial Training (CAAT)* and *Compression Scaling (CS)*. The proposed CAAT is a variation of PGD-based

---

**Algorithm 1** A Brief Overview of Our Calibration Attack Framework

---

1: **Input:** A classifier $\mathcal{F}$, input $\mathbf{x}_i$, true label $y_i$, error bound $\epsilon$, max number of attack iterations $\mathcal{I}$, attack type $\mathcal{T}$.
2: **Output:** Adversarial example $\tilde{\mathbf{x}}_i$
3: $\hat{\boldsymbol{p}}_i \leftarrow \mathcal{F}(\mathbf{x}_i); k \leftarrow \operatorname{argmax}_{j=1}^{K}(\hat{p}_{ij})$     *// k is the predicted label for the original (unattacked) input $\mathbf{x}_i$.*
4: **if** $\mathcal{T} = $ UCA or $\mathcal{T} = $ OCA **then**
5:     $\mathcal{T}' = \mathcal{T}$
6: **else if** $\mathcal{T} = $ MMA and $y_i \neq k$ **then**     *// MMA combines UCA and OCA.*
7:     $\mathcal{T}' = $ OCA
8: **else if** $\mathcal{T} = $ MMA and $y_i = k$ **then**
9:     $\mathcal{T}' = $ UCA
10: **else if** $\mathcal{T} = $ RCA **then**     *// RCA also considers both UCA and OCA.*
11:     $g \leftarrow random(1/K, 1.0);$     *// Get a random number in the range $[1/K, 1.0]$.*
12:     **if** $g > \hat{p}_k(\mathbf{x}_i)$ **then**
13:        $\mathcal{T}' = $ OCA
14:     **else if** $g < \hat{p}_k(\mathbf{x}_i)$ **then**
15:        $\mathcal{T}' = $ UCA
16:     **end**
17: **end**
18: $\tilde{\mathbf{x}}_i \leftarrow \mathbf{x}_i; l_{old} \leftarrow \mathcal{M}_{\mathcal{T}'}(\tilde{\mathbf{x}}_i, k)$
19: **for** $i = 1$ **to** $\mathcal{I}$ **do**
20:     $\boldsymbol{\delta} = FindPerturb\,(\tilde{\mathbf{x}}_i, \epsilon)$ *// Find perturbation based on attack algorithm (e.g., Square Attack) and bound.*
21:     $\tilde{\mathbf{x}}_{new} \leftarrow \tilde{\mathbf{x}}_i + \boldsymbol{\delta};$
22:     $\hat{\boldsymbol{p}}_i \leftarrow \mathcal{F}(\tilde{\mathbf{x}}_{new}); k_{new} \leftarrow \operatorname{argmax}_{j=1}^{K}(\hat{p}_{ij})$
23:     $l_{new} \leftarrow \mathcal{M}_{\mathcal{T}'}(\tilde{\mathbf{x}}_{new}, k)$
24:     **if** $(l_{new} < l_{old}$ and $k_{new} = k)$ **then**
25:        $\tilde{\mathbf{x}}_i \leftarrow \tilde{\mathbf{x}}_{new}; l_{old} \leftarrow l_{new}$
26:     **end**
27:     **if** $(l_{old} < 0.01)$ or $(\mathcal{T} = $ RCA and $\hat{p}_k(\tilde{\mathbf{x}}_i) = g)$ **then**
28:        Break the *for* loop
29:     **end**
30: **end for**

---

adversarial training that utilizes our white-box calibration attacks to generate adversarial training examples for each minibatch during training. Hence, both under- and overconfident examples with the model's original predicted label preserved are exclusively used to train the model.

Our proposed CS defence is a post-process scaling technique, based on the assumption that an effective classifier often has a high level of accuracy and confidence, so calibration attacks typically cause the most harm by lowering the confidence scores. CS hence aims to scale low confidence scores to high values to mitigate the damage of miscalibration. Specifically, the range of confidence is split into $B_M$ equally sized bins. Datapoints in each bin $b_m \in \{1, ..., B_M\}$ are mapped to a new bin that has higher confidences in a more compressed range. For a datapoint $\mathbf{x}_i$ with the output-layer logits $\{r_1, ..., r_K\}$ and the probability $\hat{p}_k(\mathbf{x}_i)$ for the predicated class $k$, a temperature $T$ is found to obtain a newly predicted probability $\hat{p}_{new} = \arg\max_i \left(\frac{\exp(r_i/T)}{\sum_j exp(r_j)/T}\right)$ so that $\hat{p}_{new} = \operatorname{minconf}(b'_m) + \frac{\hat{p}_k - \operatorname{minconf}(b_m)}{\operatorname{range}(b_m)} * \operatorname{range}(b'_m)$, where $\operatorname{minconf}(.)$ is the minimum confidence level of a bin, and $\operatorname{range}(.)$ represents the range of the confidence (i.e., the maximum level of confidence of that bin minus the minimum). $b'$ is the bin that the original $b$ is mapped to using the corresponding $T$. We will demonstrate that even with defence, calibration attacks are still highly effective.

### 3.4 Discussion on the Importance of Remaining Well Calibrated Under the Attacks

In addition to the discussion of the technical details of calibration attacks for the victim models and defences, it is important to reemphasize why maintaining good calibration under such attacks is critical. Well-calibrated models are a crucial component for trustworthy complex systems. In terms of specific real-world applications, skewed confidence scores or uncertainties have been shown to have dramatic impacts on downstream tasks and human decision-making, and previous studies such as those of Penso et al. (2024); Jiang et al. (2012); Kompa et al. (2021) have shown the effect of miscalibration on down-stream authority mechanisms in the context of areas like medical imaging. Specific use-cases can also be found in child maltreatment hotline screening models De-Arteaga et al. (2020), legal domain Bernsohn et al. (2024), and anomalous behaviour detectors Shashanka et al. (2016) where heavily miscalibrated outputs present a failure point to methods.

Moreover, prior works (Zhang et al., 2020; Rechkemmer & Yin, 2022) have examined the role of skewed confidence affecting the level of trust humans have in a system, which influences applications where humans and AI decision-making occurs symbiotically. These works demonstrate that when models produce high confidence scores, users are more prone to rely on the AI for decision-making. Dhuliawala et al. (2023) conduct user studies that show that confidently incorrect predictions lead to over reliance and heavily undermine user trust in human-AI collaboration settings, requiring only a few incorrect instances with inaccurate confidence estimates to damage user trust and performance. Please refer to Appendix B for detailed discussion on downstream impacts.

## 4 Experiments

### 4.1 Experimental Setup

**Implementation Details.** As discussed earlier, calibration attacks can be built on different adversarial attack frameworks. In our implementation, we use *Square Attack* (SA) (Andriushchenko et al., 2020), which is one of the most popular black-box approaches and is highly effective. SA achieves state-of-the-art (SOTA) performance in terms of query efficiency and success rate, even outperforming some white-box methods. Our white-box calibration attacks are based on the popular Projected Gradient Descent (PGD) framework (Madry et al., 2018). The implementation details such as hyperparameters are discussed in Appendix D.

**Models.** We include both convolutional (ResNet (He et al., 2016)) and non-convolutional attention-based models (Vision Transformer (ViT) (Dosovitskiy et al., 2021)) in our study. Details can be found in Appendices D and E.

**Datasets.** We performed a comprehensive study on CIFAR-100 (Krizhevsky & Hinton, 2009) and Caltech-101 (Fei-Fei et al., 2004). We also included the German Traffic Sign Recognition Benchmark (GTSRB) (Houben et al., 2013) for its direct implication for safety.

**Metrics.** Two calibration metrics are used in our experiments: the standard Expected Calibration Error (ECE) (Pakdaman Naeini et al., 2015) and the recent Kolmogorov-Smirnov Calibration Error (KS error) (Gupta et al., 2021). In addition, we evaluate attacks' efficiency using the average and median number of queries for the attack to complete (Andriushchenko et al., 2020). Average confidence of predictions is also leveraged to judge the degree that the confidence scores are affected. (See Appendix D for details.)

Same as in previous works (Guo et al., 2017; Kumar et al., 2018; Minderer et al., 2021), we focus on intrinsic evaluation metrics in this research, which isolate our analysis from downstream tasks to abstain from task-specific conclusions, particularly considering the downstream tasks here are very diverse. In addition, the harm from miscalibration in downstream applications has been well studied in prior work, as detailed in Appendix B.

**Detailed Attack Settings.** Descriptions of other attack settings (e.g., $l_\infty$, $l_2$ and iterations) are in Appendix D and E.

## 4.2 Overall Performance

Table 1 depicts the overall performance of black-box attacks under the $l_\infty$ norm. Additional results of $l_2$ attacks are included in Table 9 in Appendix H.10. The experiments show that the attacks are highly effective. UCA, MMA and RCA can bring ECE and KS to very high values. OCA can successfully raise average confidences (the last column in Table 1) to extremely high levels (in many cases, close to 100%), but given the original high accuracy of the victim models, increasing confidence levels will not have a drastic effect on calibration error. (OCA will be more harmful on less accurate models.) For the MMA, the theoretical highest levels of miscalibration are not reached due to the limited number of iterations that we run the attacks. Regarding different architectures, the attention-based ViT models are seen to be more miscalibrated compared to the convolution-based ResNet models. The white-box results are included in Appendix H.1. While ResNet and ViT are the most representative frameworks for convolution and attention-based models respectively, in Appendix H.8 we include additional experiments on the Swin Transformer (Liu et al., 2021), which is a famous extension of ViT. The results show similar trends as those presented in Table 1. Again, Figure 1 in the introduction section demonstrates the attack effects using calibration diagrams, visually showing the severity of miscalibration.

The general trend of the $l_2$ calibration attacks (Appendix H.10) is similar to that of the $l_\infty$ attacks, but the latter are found to be more effective in generating more significant miscalibration. Hence, in the remainder of the paper, we focus on the $l_\infty$ attacks unless otherwise specified.

Table 1: Results of underconfidence, overconfidence, maximum miscalibration, and random confidence attack. Accuracy of victim models are included.

| | Avg #q | Med. #q | ECE | KS | Avg. Conf. |
|---|---|---|---|---|---|
| **ResNet** | | | | | |
| **CIFAR-100** (Accuracy: $0.881_{\pm0.002}$) | | | | | |
| Pre-atk | - | - | $.052_{\pm.006}$ | $.035_{\pm.006}$ | $.916_{\pm.006}$ |
| UCA | $74.3_{\pm3.4}$ | $42.7_{\pm1.5}$ | $.540_{\pm.005}$ | $.479_{\pm.001}$ | $.465_{\pm.005}$ |
| OCA | $16.0_{\pm0.8}$ | $1.0_{\pm0.0}$ | $.124_{\pm.002}$ | $.124_{\pm.002}$ | $.996_{\pm.000}$ |
| MMA | $72.9_{\pm2.8}$ | $41.5_{\pm2.8}$ | $.606_{\pm.002}$ | $.497_{\pm.002}$ | $.502_{\pm.002}$ |
| RCA | $68.9_{\pm4.6}$ | $42.7_{\pm1.2}$ | $.558_{\pm.011}$ | $.461_{\pm.003}$ | $.514_{\pm.003}$ |
| **Caltech-101** (Accuracy: $0.966_{\pm0.004}$) | | | | | |
| Pre-atk | - | - | $.035_{\pm.003}$ | $.031_{\pm.004}$ | $.936_{\pm.001}$ |
| UCA | $333.8_{\pm13.8}$ | $259.7_{\pm17.4}$ | $.361_{\pm.005}$ | $.362_{\pm.005}$ | $.605_{\pm.006}$ |
| OCA | $75.7_{\pm9.3}$ | $1.0_{\pm0.0}$ | $.028_{\pm.003}$ | $.028_{\pm.004}$ | $.992_{\pm.000}$ |
| MMA | $182.6_{\pm5.6}$ | $286.5_{\pm16.1}$ | $.397_{\pm.008}$ | $.379_{\pm.007}$ | $.618_{\pm.005}$ |
| RCA | $178.5_{\pm14.9}$ | $289.3_{\pm8.1}$ | $.344_{\pm.014}$ | $.342_{\pm.010}$ | $.638_{\pm.006}$ |
| **GTSRB** (Accuracy: $0.972_{\pm0.000}$) | | | | | |
| Pre-atk | - | - | $.019_{\pm.006}$ | $.008_{\pm.002}$ | $.980_{\pm.002}$ |
| UCA | $197.5_{\pm10.3}$ | $103.0_{\pm7.3}$ | $.396_{\pm.017}$ | $.390_{\pm.013}$ | $.591_{\pm.014}$ |
| OCA | $12.1_{\pm1.3}$ | $1.0_{\pm0.0}$ | $.029_{\pm.008}$ | $.029_{\pm.008}$ | $.998_{\pm.000}$ |
| MMA | $142.1_{\pm6.0}$ | $102.2_{\pm3.6}$ | $.419_{\pm.009}$ | $.402_{\pm.012}$ | $.597_{\pm.011}$ |
| RCA | $139.4_{\pm1.5}$ | $104.7_{\pm3.5}$ | $.399_{\pm.009}$ | $.386_{\pm.005}$ | $.599_{\pm.007}$ |
| **ViT** | | | | | |
| | Avg #q | Med. #q | ECE | KS | Avg. Conf. |
| **CIFAR-100** (Accuracy: $0.935_{\pm0.002}$) | | | | | |
| Pre-atk | - | - | $.064_{\pm.006}$ | $.054_{\pm.005}$ | $.882_{\pm.004}$ |
| UCA | $118.5_{\pm2.4}$ | $62.0_{\pm3.1}$ | $.572_{\pm.007}$ | $.553_{\pm.004}$ | $.404_{\pm.003}$ |
| OCA | $524.7_{\pm88.7}$ | $510.5_{\pm114.3}$ | $.043_{\pm.007}$ | $.043_{\pm.006}$ | $.974_{\pm.001}$ |
| MMA | $104.8_{\pm7.5}$ | $62.7_{\pm4.7}$ | $.616_{\pm.003}$ | $.564_{\pm.000}$ | $.431_{\pm.001}$ |
| RCA | $106.4_{\pm3.0}$ | $70.3_{\pm1.5}$ | $.549_{\pm.002}$ | $.505_{\pm.003}$ | $.471_{\pm.007}$ |
| **Caltech-101** (Accuracy: $0.961_{\pm0.024}$) | | | | | |
| Pre-atk | - | - | $.137_{\pm.059}$ | $.136_{\pm.060}$ | $.825_{\pm.083}$ |
| UCA | $325.5_{\pm16.7}$ | $273.7_{\pm23.7}$ | $.426_{\pm.044}$ | $.426_{\pm.044}$ | $.536_{\pm.068}$ |
| OCA | $52.1_{\pm40.9}$ | $1.0_{\pm0.0}$ | $.081_{\pm.042}$ | $.079_{\pm.040}$ | $.881_{\pm.067}$ |
| MMA | $150.7_{\pm12.1}$ | $269.7_{\pm25.1}$ | $.415_{\pm.036}$ | $.414_{\pm.034}$ | $.551_{\pm.058}$ |
| RCA | $129.0_{\pm17.3}$ | $315.0_{\pm17.4}$ | $.364_{\pm.016}$ | $.364_{\pm.016}$ | $.598_{\pm.040}$ |
| **GTSRB** (Accuracy: $0.947_{\pm0.006}$) | | | | | |
| Pre-atk | - | - | $.040_{\pm.005}$ | $.026_{\pm.017}$ | $.922_{\pm.024}$ |
| UCA | $169.8_{\pm15.0}$ | $88.3_{\pm6.7}$ | $.459_{\pm.015}$ | $.452_{\pm.019}$ | $.498_{\pm.026}$ |
| OCA | $94.9_{\pm45.9}$ | $3.7_{\pm4.6}$ | $.029_{\pm.003}$ | $.030_{\pm.004}$ | $.976_{\pm.011}$ |
| MMA | $137.1_{\pm4.3}$ | $88.3_{\pm6.7}$ | $.519_{\pm.020}$ | $.480_{\pm.020}$ | $.509_{\pm.024}$ |
| RCA | $129.5_{\pm7.4}$ | $97.2_{\pm9.9}$ | $.454_{\pm.012}$ | $.432_{\pm.016}$ | $.538_{\pm.019}$ |

Overall, calibration attacks are generating severe miscalibration, which, compared to the pre-attack values, can increase ECE and KS by over 10 times in many cases. Calibration attacks are very effective without changing the prediction accuracy, which could raise serious concerns for any down-stream applications relying on confidence.

## 4.3 Detection Difficulty Analysis

This section shows calibration attacks are also difficult to detect. To investigate this, we run the popular adversarial attack detection methods on the attacks against ResNet-50: Local Intrinsic Dimensionality (LID) (Ma et al., 2018), Mahalanobis Distance (MD) (Lee et al., 2018), and SpectralDefense (Harder et al., 2021). The details behind the settings for each detection method can be found in Appendix G. Table 2 depicts the main results of the effectiveness of the detectors in terms of Area Under the Curve (AUC) and Detection Accuracy, under different types of calibration attacks and using both the white-box and black-box approaches. We can see that there are consistent decreases in detection performances, particularly in SA,

Table 2: Attack detection results comparing original version of SA and PGD attacks with their calibration attack counterparts.

| | SA | | SA-UCA | | SA-OCA | | SA-MMA | | PGD | | PGD-UCA | | PGD-OCA | | PGD-MMA | |
|---|---|---|---|---|---|---|---|---|---|---|---|---|---|---|---|---|
| | AUC | Acc | AUC | Acc | AUC | Acc | AUC | Acc | AUC | Acc | AUC | Acc | AUC | Acc | AUC | Acc |
| **CIFAR-100** | | | | | | | | | | | | | | | | |
| LID | 90.1 | 82.9 | 54.2 | 54.3 | 63.9 | 61.6 | 54.5 | 54.0 | 93.7 | 87.7 | 64.5 | 67.1 | 88.7 | 84.3 | 63.3 | 66.0 |
| MD | 99.8 | 98.9 | 90.2 | 80.5 | 78.1 | 74.8 | 89.4 | 79.9 | 99.3 | 98.5 | 83.1 | 74.4 | 96.7 | 93.8 | 81.9 | 73.3 |
| Spect. | 100 | 98.0 | 70.5 | 65.5 | 52.3 | 50.5 | 71.6 | 65.5 | 100 | 100 | 74.9 | 67.0 | 94.2 | 90.0 | 64.8 | 62.5 |
| **Caltech-101** | | | | | | | | | | | | | | | | |
| LID | 70.2 | 64.4 | 62.5 | 61.1 | 65.3 | 62.3 | 58.6 | 59.5 | 84.8 | 78.1 | 53.6 | 54.8 | 89.1 | 81.9 | 53.9 | 55.1 |
| MD | 88.9 | 81.3 | 81.9 | 74.7 | 81.8 | 70.1 | 81.8 | 74.5 | 91.6 | 84.6 | 60.1 | 57.9 | 90.6 | 84.7 | 62.6 | 60.3 |
| Spect. | 98.0 | 94.5 | 59.1 | 53.0 | 51.8 | 50.0 | 56.9 | 53.0 | 93.4 | 88.5 | 67.8 | 66.5 | 93.7 | 90.5 | 64.2 | 62.0 |
| **GTSRB** | | | | | | | | | | | | | | | | |
| LID | 86.3 | 77.1 | 71.4 | 68.1 | 72.5 | 69.8 | 72.4 | 66.7 | 95.7 | 89.1 | 88.8 | 86.3 | 94.6 | 87.4 | 87.0 | 85.3 |
| MD | 95.5 | 89.6 | 83.7 | 77.8 | 74.4 | 74.9 | 85.6 | 79.0 | 100 | 99.8 | 94.6 | 93.6 | 97.9 | 97.7 | 92.9 | 92.6 |
| Spect. | 99.1 | 98.0 | 83.0 | 79.0 | 50.4 | 50.5 | 83.1 | 79.0 | 99.4 | 98.5 | 94.8 | 93.0 | 99.0 | 99.9 | 99.0 | 99.9 |

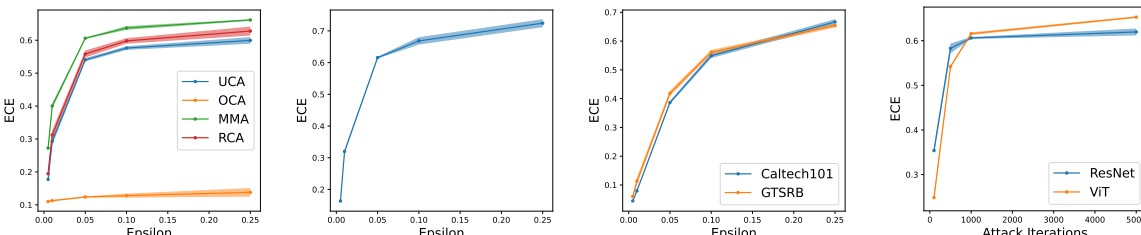

Figure 2: The influence of perturbation noise levels $\epsilon$ (the left three sub-figures) and attack iterations (the right sub-figure). Sub-figure-1 (the left most) presents the comparison between the ECE scores of the different calibration attacks at different $\epsilon$ values using ResNet-50 models trained on CIFAR-100. Sub-figure-2: ECE vs. $\epsilon$ using maximum miscalibration attacks on ViT models trained on CIFAR-100. Sub-figure-3: ECE vs. $\epsilon$ using maximum miscalibration attacks on the ResNet-50 models trained on Caltech-101 and GTSRB. Sub-figure-4: Effect of the numbers of attack iterations on the ability of the attack algorithm. The first three sub-figures are created at the $1000^{th}$ attack iteration.

where the decreases are often more than 20%. The existing detection methods are shown to be less reliable for calibration attacks.

## 4.4 Insights on Key Aspects of Attacks

We analyze key aspects of calibration attacks: attack directions, noise bounds, and attack iterations under the $l_\infty$-based attacks. More details can be found in Appendix H, including attack effectiveness on data with varying imbalance ratios.

**Comparison of Efficiency of Underconfidence vs. Overconfidence Attacks.** The two directions of calibration attacks, underconfidence or overconfidence, is a basic building block for constructing the four forms of calibration attacks. We perform further studies to understand which direction is most efficient. To the end, we identify all data points in the test set that are around certain predefined base confidence levels. In this study, we choose two base confidences: 80% and 90%. All the test cases that have confidences within 1% around these two base confidences are included in this experiment. We attack these examples by making the victim models produce either a 10% increase or decrease in confidence. In Table 3 we can see a consistent pattern — for both base confidence levels, it takes notably fewer queries to create underconfidence than overconfidence adversarial examples. The former attack is also more effective in affecting the average confidences. This property could be further utilized to design calibration attacks (e.g., under the circumstances where computing resources or attack latency are concerned.)

**Perturbation Noise Levels.** We study the effect of perturbation noise levels and how low the value could be in order to construct notable harm. As shown in the left three sub-figures in Figure 2, under different setups, the attacks are highly successful even at a low level of noise (e.g., $\epsilon = 0.05$ or even lower). We can also see the rise in ECE is sharp when $\epsilon$ increases, and it plateaus quickly.

**Iterations.** The numbers of iterations can help measure the efficiency and cost of attacks (in terms of both time and computing expenditure). The last sub-figure in Figure 2 shows ECE vs. iteration numbers when

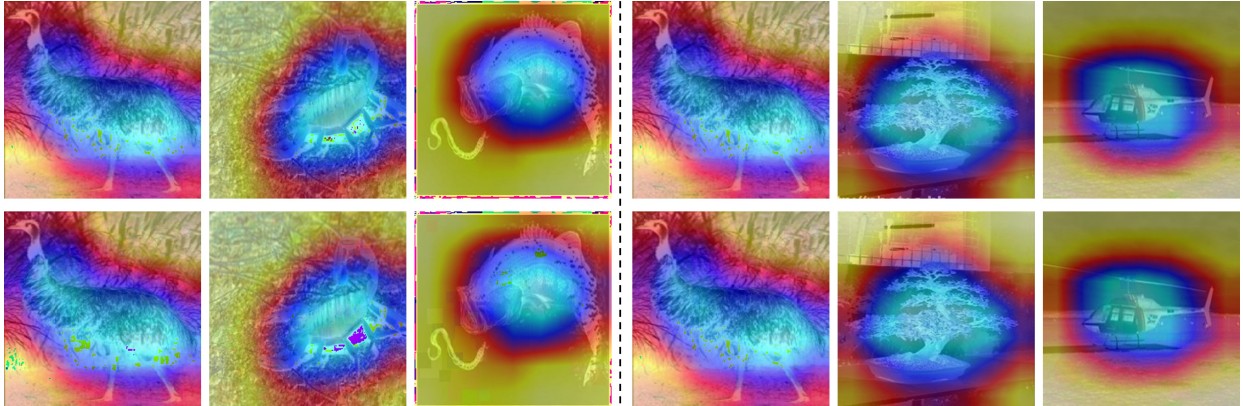

Figure 3: The GradCAM visualizations shows the image regions most responsible for the decisions of ResNet-50 before (top row) and after (bottom row) attacks. The left three images are under `UCA` and the right three the `OCA`.

applying `MMA` to the victim models on CIFAR-100. In Appendix H.3, we provide more detailed comparison, showing that the calibration attacks consistently produced a higher degree of miscalibration compared to the original unaltered SA. ECE begins to saturate at 500 iterations, but even at 100 iterations the victim models become heavily miscalibrated.

## 4.5 Qualitative Analysis

**GradCAM Visualization.** In Figure 3 we show the coarse localization maps produced with GradCAM (Selvaraju et al., 2017), which highlights the most important regions that a model relies on to make prediction using the gradients from different layers of a network. We apply GradCAM to our ResNet-50 models that are fine-tuned and tested on Caltech-101, based on the standard attack settings discussed in Section D. We choose images where the attacks led to large change in predicted confidence (at least 10%). Figure 3 shows several representative images. We can see that the coarse localization maps have minimal to no noticeable changes after the adversarial images are produced, especially in the case of the overconfidence attacked images. This analysis shows that it could be difficult to identify the attacks based on gradient visualization methods.

**t-SNE.** Appendix H.4 provides detailed t-SNE visualization on the effect of different forms of calibration attacks. The visualization shows that the overconfidence attack causes the representations for different classes to be split apart as much as possible, while the underconfidence attack causes a more jumbled representation with most data points falling closely to the decision boundary. The visualization shows that the attacks achieve their intended goals.

Table 3: Comparison between the efficiency of the `UCA` and `OCA`.

| ResNet | | | | |
|---|---|---|---|---|
| | # datapoints | Avg. #q | Med. #q | Avg. Conf. |
| **CIFAR-100** | | | | |
| 90% -10% UCA | $186.0_{\pm4.4}$ | $8.5_{\pm0.5}$ | $5.7_{\pm0.6}$ | $77.9_{\pm1.4}$ |
| 90% +10% OCA | $186.0_{\pm4.4}$ | $36.5_{\pm1.5}$ | $31.0_{\pm1.7}$ | $99.0_{\pm0.0}$ |
| 80% -10% UCA | $94.0_{\pm4.2}$ | $6.5_{\pm0.5}$ | $4.3_{\pm0.7}$ | $67.9_{\pm0.4}$ |
| 80% +10% OCA | $94.0_{\pm4.2}$ | $9.8_{\pm1.7}$ | $7.0_{\pm1.4}$ | $91.3_{\pm0.1}$ |
| **Caltech-101** | | | | |
| 90% -10% UCA | $160.7_{\pm7.8}$ | $39.4_{\pm11.8}$ | $31.7_{\pm8.5}$ | $80.2_{\pm0.0}$ |
| 90% +10% OCA | $160.7_{\pm7.8}$ | $225.6_{\pm55.5}$ | $212.8_{\pm63.3}$ | $98.8_{\pm0.2}$ |
| 80% -10% UCA | $53.3_{\pm24.0}$ | $17.5_{\pm12.1}$ | $16.0_{\pm19.1}$ | $63.1_{\pm2.1}$ |
| 80% +10% OCA | $53.3_{\pm24.0}$ | $30.0_{\pm10.3}$ | $21.7_{\pm11.3}$ | $89.5_{\pm0.4}$ |
| **GTSRB** | | | | |
| 90% -10% UCA | $48.0_{\pm4.2}$ | $11.7_{\pm0.1}$ | $8.2_{\pm1.4}$ | $77.5_{\pm1.2}$ |
| 90% +10% OCA | $48.0_{\pm4.2}$ | $98.5_{\pm4.0}$ | $62.8_{\pm5.7}$ | $99.1_{\pm0.0}$ |
| 80% -10% UCA | $30.0_{\pm0.7}$ | $9.2_{\pm0.9}$ | $6.0_{\pm1.1}$ | $70.5_{\pm0.7}$ |
| 80% +10% OCA | $30.0_{\pm0.7}$ | $18.6_{\pm3.4}$ | $12.8_{\pm0.4}$ | $91.0_{\pm0.1}$ |

| ViT | | | | |
|---|---|---|---|---|
| | # datapoints | Avg. #q | Med. #q | Avg. Conf. |
| **CIFAR-100** | | | | |
| 90% -10% UCA | $392.7_{\pm33.5}$ | $20.0_{\pm1.6}$ | $9.3_{\pm1.2}$ | $76.1_{\pm0.3}$ |
| 90% +10% OCA | $392.7_{\pm33.5}$ | $883.7_{\pm120.2}$ | $883.7_{\pm120.2}$ | $97.1_{\pm0.1}$ |
| 80% -10% UCA | $138.7_{\pm5.1}$ | $8.8_{\pm1.7}$ | $5.5_{\pm0.5}$ | $66.1_{\pm0.3}$ |
| 80% +10% OCA | $138.7_{\pm5.1}$ | $55.8_{\pm1.4}$ | $20.7_{\pm4.7}$ | $90.0_{\pm0.1}$ |
| **Caltech-101** | | | | |
| 90% -10% UCA | $472.0_{\pm161.0}$ | $264.7_{\pm56.6}$ | $198.0_{\pm51.0}$ | $81.1_{\pm0.5}$ |
| 90% +10% OCA | $472.0_{\pm161.0}$ | $700.7_{\pm518.5}$ | $700.7_{\pm518.5}$ | $93.2_{\pm0.2}$ |
| 80% -10% UCA | $222.0_{\pm17.3}$ | $149.5_{\pm19.3}$ | $66.5_{\pm9.8}$ | $70.0_{\pm0.6}$ |
| 80% +10% OCA | $222.0_{\pm17.3}$ | $323.9_{\pm123.5}$ | $269.3_{\pm210.7}$ | $86.0_{\pm0.2}$ |
| **GTSRB** | | | | |
| 90% -10% UCA | $184.0_{\pm27.5}$ | $31.0_{\pm6.5}$ | $12.7_{\pm1.9}$ | $76.1_{\pm0.9}$ |
| 90% +10% OCA | $184.0_{\pm27.5}$ | $235.3_{\pm69.5}$ | $180.0_{\pm92.0}$ | $96.5_{\pm0.2}$ |
| 80% -10% UCA | $79.0_{\pm28.6}$ | $16.1_{\pm6.7}$ | $7.7_{\pm2.3}$ | $67.3_{\pm0.2}$ |
| 80% +10% OCA | $79.0_{\pm28.6}$ | $140.1_{\pm63.2}$ | $41.7_{\pm43.7}$ | $89.6_{\pm0.8}$ |

## 4.6 Confidence Certification

We follow the method in Kumar et al. (2020) to compare the lower bound of expected confidence of a smoothed classifier between base images and their under- and overconfidence attacked counterparts. We find that the lower bound is not significantly different between adversarially attacked images and their originals, though the accuracy of the smoothed classifier is affected. This means that methods for determining certified lower bounds are not strongly affected by calibration attacked samples (similar to conventional adversarial samples), but limitations in efficiency and accuracy of these methods means that establishing defence method against such attacks is still paramount. A detailed discussion is in Appendix H.7.

## 4.7 Attack Transferability

Determining the transferability of calibration attacks across different model architectures is of interest since although one can effectively optimize attacks against a single model, using an ensemble model could potentially be a form of defence if transferability is limited. We investigate how our attacks transfer to different victim models. Specifically in our experiments, the attacks are constructed on CIFAR-100 images with regular settings based on ViT but used on

Table 4: Attack transfer ViT to ResNet and to ensembled victim models (with ViT as target) on CIFAR-100.

|  | ECE | KS | Avg.-Conf. |
|---|---|---|---|
| Pre-atk | $.051_{\pm.005}$ | $.035_{\pm.006}$ | $.916_{\pm.006}$ |
| UCA SQ VIT $\rightarrow$ ResNet | $.104_{\pm.003}$ | $.101_{\pm.007}$ | $.780_{\pm.012}$ |
| UCA PGD VIT$\rightarrow$ ResNet | $.196_{\pm.020}$ | $.194_{\pm.021}$ | $.729_{\pm.011}$ |
| Pre-atk | $.028_{\pm.005}$ | $.015_{\pm.008}$ | $.920_{\pm.002}$ |
| UCA SQ Ensemble | $.091_{\pm.010}$ | $.086_{\pm.008}$ | $.708_{\pm.012}$ |
| UCA PGD Ensemble | $.048_{\pm.011}$ | $.014_{\pm.006}$ | $.637_{\pm.030}$ |

(i) ResNet and (ii) ensemble victim models combining prediction of both ResNet-50 and ViT (Table 4). We focus on the transfer of the UCA attack, given that OCA has little room to skew confidence due to the already high accuracy of victim models. The attacks show transferability, although in general they become less effective when applied on different architectures. This is expected because the attacks are constructed without considering the target victim models' properties. Most notably, the simple ensemble of two typical victim models (ViT and ResNet) is still greatly subject to the attacks (the output distributions of ResNet and ViT are averaged to get the ensemble), so using ensembles as a defence does not fully protect against calibration attacks. To further explore transferability, attack methods need to be explicitly redesigned (Xie et al., 2018; Wang et al., 2021), which could be further studied in the future.

## 4.8 Influence of Model Scale on Attack Effectiveness

To help understand the influence of varying model sizes on attack performance, we perform a study on models of different sizes over different architectures across four attack variants. Specifically, for ResNet, we compare ResNet-18, ResNet-50, and ResNet-152. For ViT, we included Base (*vit-base-patch16-224-in21k*) and Large models (*vit-large-patch16-224-in21k*). The model training closely follows the settings detailed in Appendix D.1. We fine-tune the models for each variant and average the results. We ran attacks with the same settings used to produce Tables 1 and 3.

In Table 5 we present the results of the strongest variant of calibration attack (i.e., MMA). It shows that the proposed calibration attacks are effective among different model sizes.

Table 5: Attack effectiveness of MMA (black-box setup) across different model sizes on CIFAR-100.

|  | Avg #q | Med. #q | ECE | KS | Avg.Conf. |
|---|---|---|---|---|---|
| *ResNet-18 (Acc: .854$_{\pm.004}$)* | | | | | |
| Pre | - | - | $.037_{\pm.003}$ | $.016_{\pm.005}$ | $.870_{\pm.001}$ |
| Post | $52.9_{\pm1.4}$ | $36.0_{\pm0.0}$ | $.531_{\pm.009}$ | $.471_{\pm.004}$ | $.446_{\pm.003}$ |
| *ResNet-50 (Acc: .881$_{\pm.002}$)* | | | | | |
| Pre | - | - | $.052_{\pm.006}$ | $.035_{\pm.006}$ | $.916_{\pm.006}$ |
| Post | $72.9_{\pm2.8}$ | $41.5_{\pm2.8}$ | $.606_{\pm.002}$ | $.497_{\pm.002}$ | $.502_{\pm.002}$ |
| *ResNet-152 (Acc: .883$_{\pm.002}$)* | | | | | |
| Pre | - | - | $.051_{\pm.005}$ | $.046_{\pm.004}$ | $.929_{\pm.005}$ |
| Post | $80.5_{\pm3.8}$ | $47.0_{\pm3.5}$ | $.539_{\pm.002}$ | $.475_{\pm.002}$ | $.472_{\pm.008}$ |
| *ViT-base (Acc: .935$_{\pm.002}$)* | | | | | |
| Pre | - | - | $.064_{\pm.006}$ | $.054_{\pm.005}$ | $.882_{\pm.004}$ |
| Post | $104.8_{\pm7.5}$ | $62.7_{\pm4.7}$ | $.616_{\pm.003}$ | $.564_{\pm.000}$ | $.431_{\pm.001}$ |
| *ViT-large (Acc: .936$_{\pm0.001}$)* | | | | | |
| Pre | - | - | $.037_{\pm.009}$ | $.022_{\pm.012}$ | $.921_{\pm.021}$ |
| Post | $122.1_{\pm2.0}$ | $76.3_{\pm3.2}$ | $.531_{\pm.012}$ | $.508_{\pm.020}$ | $.450_{\pm.024}$ |

The model sizes do not have a major impact on the susceptibility to calibration attacks, as the overall ECE and KS scores do not follow a clear pattern along with varying model sizes. However, when models scale up, the numbers of queries increase in general, suggesting that attacking complex models requires more rounds of attacks in most cases. In Appendix H.11 we present detailed results in the black-box and white-box setups across three architectures under the four types of attacks, which agree with our aforementioned observations.

# 5 Defending Against Calibration Attacks

We compare a wide range of recalibration and defence methods under the setup of the maximum miscalibration attacks, which, as shown above, are among the most effective calibration attacks.

Specifically, for post-calibration methods, we include Temperature Scaling (TS) (Guo et al., 2017), Multi-domain Temperature Scaling (MD-TS) (Yu et al., 2022), and calibration with splines (Splines) (Gupta et al., 2021). For training-based regularization methods we include two effective models, DCA (Liang et al., 2020) and SAM (Foret et al., 2021). Regarding adversarial defence methods, we test the top-3 SOTA models under the $l_\infty$ attack for CIFAR-100, using WideResNet on the RobustBench leaderboard (Croce et al., 2021), which are only available for CIFAR-10 and CIFAR-100, hence we run over CIFAR-100 to compare with our previous baselines. We further include a recent post-process defence called Adversarial Attack Against Attackers (AAA) (Chen et al., 2022) in addition to the the most common defences in the form of PGD-based adversarial training (AT) (Madry et al., 2018). We also included the two defences proposed for calibrations attacks: CAAT and CS, which are introduced in Section 3.3. (The experiment setup is described in Appendix D, and the detailed description of these baselines are in Appendix E.2).

**Results and Analyses.** Table 6 shows the experiment results of query efficiency, accuracy, and the ECE and KS errors, before (PrECE and PrKS) and after the attacks (PsECE and PrKS), using different recalibration and defence models. The methods with the best performances in terms of post-attack ECE (PsECE) and KS (PsKS) are marked in bold, and the second best are underlined.

We can see that CS is overall the strongest methods at maintaining low calibration errors on post-attack datapoints. It showed the best post-attack calibration performance in five out of six setups on PsECE, and ranked among top 2 in all the other setups. The other top calibration performances are distributed among AAA, Splines, and AT. Know-

Table 6: Effectiveness of calibration methods and adversarial defences. The best performances of post-attack ECE (PsECE) and KS (PsKS) are marked in bold, and the second best are marked with underlines.

| WideResNet | | | | | | |
|---|---|---|---|---|---|---|
| | Avg#q | Med#q | Acc | PrECE | PsECE | PrKS | PsKS |
| **CIFAR-100** | | | | | | | |
| Gowal, '20 | 63.5 | 86.5 | .690 | .137 | .248 | .137 | .200 |
| Rebuffi, '21 | 36.4 | 51.0 | .622 | .190 | .209 | .189 | .198 |
| Pang, '22 | 50.5 | 64.5 | .638 | .185 | .214 | .187 | .195 |

| ResNet-50 | | | | | | |
|---|---|---|---|---|---|---|
| | Avg#q | Med#q | Acc | PrECE | PsECE | PrKS | PsKS |
| **CIFAR-100** | | | | | | | |
| TS | 74.1 | 40.0 | .880 | .034 | .643 | .007 | .530 |
| MD-TS | 63.6 | 44.0 | .880 | .068 | .617 | .069 | .581 |
| Splines | 6.6 | 83.0 | .876 | .020 | .681 | .019 | .573 |
| DCA | 68.0 | 39.0 | .880 | .049 | .604 | .039 | .492 |
| SAM | 83.8 | 44.5 | .882 | .033 | .609 | .014 | .506 |
| AAA | 7.7 | 31.5 | .880 | .038 | .225 | .011 | **.123** |
| AT | 65.9 | 60.0 | .790 | .035 | .431 | .022 | .279 |
| CAAT | 66.9 | 44.0 | .842 | .048 | .504 | .036 | .440 |
| CS | 64.4 | 39.0 | .880 | .051 | **.218** | .041 | .145 |
| **Caltech-101** | | | | | | | |
| TS | 194.7 | 276.0 | .970 | .014 | .347 | .005 | .322 |
| MD-TS | 178.6 | 280.0 | .970 | .025 | .319 | .017 | .321 |
| Splines | 4.5 | 150.0 | .970 | .019 | .104 | .010 | .095 |
| DCA | 189.6 | 269.0 | .962 | .038 | .418 | .027 | .392 |
| SAM | 191.1 | 276.0 | .970 | .051 | .429 | .049 | .414 |
| AAA | 1.1 | 20.0 | .964 | .061 | .100 | .058 | .085 |
| AT | 23.7 | 194.0 | .918 | .038 | .079 | .018 | .068 |
| CAAT | 127.5 | 206.0 | .972 | .017 | .264 | .012 | .266 |
| CS | 179.4 | 254.0 | .970 | .026 | **.065** | .017 | **.067** |
| **GTSRB** | | | | | | | |
| TS | 160.6 | 111.0 | .972 | .019 | .396 | .018 | .377 |
| MD-TS | 146.5 | 110.5 | .972 | .028 | .468 | .023 | .469 |
| Splines | 0.7 | 22.0 | .972 | .018 | .129 | .007 | .123 |
| DCA | 130.2 | 97.0 | .976 | .017 | .389 | .011 | .372 |
| SAM | 117.4 | 87.0 | .978 | .012 | .384 | .003 | .371 |
| AAA | 3.1 | 51.0 | .972 | .023 | **.071** | .014 | **.059** |
| AT | 37.7 | 117.5 | .962 | .017 | .160 | .007 | .135 |
| CAAT | 121.7 | 115.0 | .968 | .020 | .324 | .017 | .317 |
| CS | 151.2 | 111.0 | .972 | .019 | .097 | .020 | .095 |

| ViT | | | | | | |
|---|---|---|---|---|---|---|
| | Avg#q | Med#q | Acc | PrECE | PsECE | PrKS | PsKS |
| **CIFAR-100** | | | | | | | |
| TS | 117.7 | 73.0 | .938 | .014 | .568 | .010 | .515 |
| MD-TS | 97.1 | 59.0 | .938 | .026 | .542 | .021 | .514 |
| Splines | 9.9 | 130.5 | .938 | .023 | .405 | .016 | .358 |
| DCA | 125.6 | 69.0 | .944 | .024 | .565 | .011 | .519 |
| SAM | 123.0 | 66.0 | .942 | .072 | .607 | .064 | .561 |
| AAA | 0.8 | 42.0 | .938 | .106 | .200 | .092 | .161 |
| AT | 86.8 | 77.0 | .886 | .066 | .519 | .063 | .439 |
| CAAT | 102.4 | 56.0 | .922 | .026 | .537 | .010 | .506 |
| CS | 97.0 | 59.0 | .938 | .044 | **.137** | .033 | **.142** |
| **Caltech-101** | | | | | | | |
| TS | 154.8 | 280.0 | .972 | .030 | .313 | .023 | .264 |
| MD-TS | 140.1 | 272.0 | .938 | .038 | .272 | .012 | .253 |
| Splines | 0.5 | 45.0 | .972 | .035 | .071 | .017 | **.049** |
| DCA | 140.9 | 254.5 | .976 | .039 | .345 | .025 | .345 |
| SAM | 145.9 | 278.0 | .962 | .170 | .459 | .170 | .459 |
| AAA | 0.3 | 20.5 | .972 | .189 | .196 | .189 | .198 |
| AT | 48.2 | 188.0 | .946 | .132 | .229 | .132 | .231 |
| CAAT | 136.2 | 341.0 | .986 | .048 | .316 | .049 | .318 |
| CS | 143.3 | 277.0 | .934 | .025 | **.068** | .018 | .068 |
| **GTSRB** | | | | | | | |
| TS | 132.9 | 80.0 | .950 | .038 | .463 | .033 | .410 |
| MD-TS | 130.5 | 81.5 | .940 | .017 | .436 | .012 | .422 |
| Splines | 1.7 | 16.0 | .950 | .040 | .115 | .041 | **.067** |
| DCA | 132.8 | 92.0 | .950 | .052 | .506 | .037 | .476 |
| SAM | 133.9 | 103.0 | .944 | .070 | .505 | .069 | .473 |
| AAA | 0.1 | 4.0 | .950 | .053 | .128 | .049 | .110 |
| AT | 66.9 | 124.0 | .930 | .132 | .320 | .130 | .317 |
| CAAT | 118.9 | 85.0 | .932 | .066 | .446 | .055 | .431 |
| CS | 130.5 | 81.5 | .950 | .027 | **.092** | .035 | .089 |

ing the property of calibration attacks and organizing defences accordingly is helpful to ensure better defence results.

Overall, from the ECE and KS scores, we can see that there are still significant limitations on recalibration and defences for calibration attacks, which invites more future research. Simple and widely used calibration models like TS are effective on clean data prior to the attacks, but they offer very little benefit post-attack. Training-based models like DCA and SAM also tend to bring few benefits after being attacked — the post-attack ECE and KS errors are not substantially different compared to the vanilla models. Lastly, we can conclude that although the defence methods from the leaderboard are generally the most adversarially resistant, their inherent high levels of miscalibration even before the attacks render them unsuitable for calibration-sensitive tasks.

## 6 Conclusions

We highlight and perform a comprehensive and dedicated study on calibration attacks, which aim to trap victim models into being heavily miscalibrated, hence endangering the trustworthiness of the models and any follow-up decision-making processes based on confidences. We propose four forms of calibration attacks and demonstrate their severity from different perspectives. We also show calibration attacks are difficult to detect compared to standard attacks. An investigation was conducted to study the effectiveness of a wide range of adversarial defences and calibration methods, including the defences that are specifically designed for calibration attacks. From the ECE and KS scores, we can see that there are still limitations on these recalibration and defences in handling calibration attacks. We hope this paper helps attract more attention to the attacks against confidence and hence mitigate their potential harm. We provides detailed analyses to help understand the characteristics of the attacks for future work.

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

**Appendix**

## A    Detailed Summary of Related Work

In the field of calibration, a great deal of current research is devoted to the creation of new calibration methods that can be applied to create better calibrated models while possessing as minimum overhead in applying them as possible. Methods are generally divided into either post-calibration or training-based. Post-calibration methods can be applied directly to the predictions of fully trained models at test time, and methods of this class include temperature scaling (Guo et al., 2017). More traditional methods of this type include Platt scaling (Platt, 1999), isotonic regression (Zadrozny & Elkan, 2002), and histogram binning (Zadrozny & Elkan, 2001). All of these three methods are originally formulated for binary classification settings, and work by creating a function that maps predicted probabilities based on their values more in tune with the model's level of performance. Although they are easy to apply, they often come with the limitation of needing a large degree of validation data to tune, especially with isotonic regression, and performance can struggle when applied to more out of distribution data.

The second class of methods are training-based methods, which typically add a bias during training to ensure that a model learns to become better calibrated. Often times these methods help by acting as a form of regularization that can punish high levels of overconfidence late into training. In computer vision, Mixup (Zhang et al., 2018) is a commonly used method of this type that serves as an effective regularizer by convexly combining random pairs of images and their labels and helps calibration primarily due to the use of soft, interpolated labels (Thulasidasan et al., 2019). Other methods work by adding a penalty to the loss function, like in the case of MMCE, an RKHS kernel-based measure of calibration that is added as a penalty on top of the regular loss during training so that both are optimized jointly (Kumar et al., 2018). Similarly, Tomani & Buettner (2021) create a new loss term called adversarial calibration loss that directly minimizes calibration error using adversarial examples. Given the effectiveness of many of these methods in regular testing scenarios, we desire to illustrate how well a diverse range of these methods can cope against attacks targeting model calibration and whether they possess limitations that require them to be overhauled to deal with an attack scenario.

With respect to adversarial attacks, attacks in this field are wide ranging. Well known white-box attacks include the basic Fast Gradient Sign Method (FGSM) (Goodfellow et al., 2015). This method works by finding adjustments to the input data that maximizes the loss function, and uses the backpropagated gradients to produce the adversarial examples. Projected gradient descent (PGD) (Madry et al., 2018) is popular iterative-based method that similarly uses gradient information, and has been shown to be a universal first-order adversary, and thus is the strongest form of attacks making using of gradient and loss information. In the black-box space of attacks, ZOO (Chen et al., 2017) is an example of a popular score-based attack that uses zeroth order stochastic coordinate descent to attack the model, and avoids training a substitute model. The authors make use of attack-space dimension reduction, hierarchical attacks and importance sampling to make the attack more query efficient, which is required as black-box attacks generally need a lot of queries to run compared to white-box methods.

A broad range of defences against adversarial attacks have been developed, but among the most popular and effective is adversarial training (Goodfellow et al., 2015), where during training the loss is minimized over one of or both clean and generated adversarial examples. Adversarial training however greatly increases training time due to the need to fabricate adversarial examples for every batch. Gradient masking (Carlini & Wagner, 2017) is a simple defence based on obfuscating gradients so that attacks cannot make use of gradient information to create adversarial examples, although it can easily be circumvented in many cases for white-box models (Athalye et al., 2018), and black box attacks do not need gradient information in the first place. It is by and large difficult for adversarial defences to keep pace with the broad range of attacks and to be provably robust against a large number of them. Although the main topic of this work is calibration, we do focus on modelling adversarial defences and their effectiveness against these attacks.

## B   Miscalibration Effects on Downstream Tasks

In addition to the discussion of the technical details of calibration attacks for the victim models and defences, it is important to reemphasize why maintaining good calibration under such attacks is critical. Well-calibrated models are a crucial component for trustworthy complex systems. As an example, in real-life deployment, proper confidence scores are essential for determining the instances that need further examination by authority mechanisms such as an expert or a committee of them. Underconfidence attacks could cause additional strain on an authority system or any downstream processes, creating a risk of significantly slowed-down decision-making due to an increased load of cases to be examined. As another example, by attacking misclassified datapoints with an overconfidence approach, test cases may be erroneously missed by downstream systems such as braking in an autonomous vehicle before a stop sign.

In terms of specific real-world applications, skewed confidence scores or uncertainties have been shown to have dramatic impacts on downstream tasks and human decision-making, which many previous works have highlighted. Previous studies such as those of Penso et al. (2024); Jiang et al. (2012); Kompa et al. (2021) have shown the effect of miscalibration on down-stream authority mechanisms. Penso et al. (2024) discuss how medical imaging, if skipping expert review for the confident but incorrect model predictions, can have disastrous consequences, stressing the importance of being well-calibrated on incorrect predictions. Kompa et al. (2021) advocate for medical AI models being able to abstain from providing a diagnosis when there is a significant uncertainty for a patient, and to defer to additional human expertise to make a more informed diagnosis. This relies on a model that can give well-calibrated estimates.

Moreover, prior works have examined the role of skewed confidence affecting the level of trust humans have in a system, which influences applications where humans and AI decision-making occurs symbiotically Zhang et al. (2020); Rechkemmer & Yin (2022). Antifakos et al. (2005), through a study on context-aware mobile phones, show how providing explicit confidence improves user trust in a system. Zhang et al. (2020) demonstrate when a model produces high confidence scores, users are more prone to rely on the AI for decision-making. These factors show why miscalibrated outputs can have great influence on downstream processing and authority mechanisms.

As more specific examples in the healthcare domain, chest abnormality detection Dyer et al. (2021) tag radiographs with high-confidence normality to help radiologists make decisions. In child maltreatment hotlines, screening models De-Arteaga et al. (2020) are deployed where confidence is provided to suggest the likelihood that a child on the referral will experience adverse welfare-related outcomes. Call workers use confidence, along with other information, to make their decisions. The study discusses how the tool could be "glitching" out by providing incorrect confidence, compromising operability. In the legal domain, LegalLens Bernsohn et al. (2024) detects legal violations. Entities with high confidence are further processed with downstream models. In enterprise security, anomalous behaviour detectors Shashanka et al. (2016) use confidence to send alerts about likely malicious activity.

Dhuliawala et al. (2023) conduct user studies that show that confidently incorrect predictions heavily undermine user trust in human-AI collaboration settings, requiring only a few incorrect instances with inaccurate confidence estimates to damage user trust and performance, creating long-term effects with recovery being very difficult once humans have been exposed to miscalibrated confidence estimates. They observe that both confidently incorrect stimuli and unconfidently correct stimuli reduce trust, with the former being more significant, which justifies our formulation of `MMA` as having a high potential for creating mistrust by optimizing the creation of both of these stimuli.

## C   Details for Maximum Miscalibration Attacks

Proof of Proposition 3.1.

*Proof.* Let a classifier have non-zero accuracy. We cannot expect to reach the error of 100% since $p_k = 0$ cannot be the case (for the top predicted class) nor can $\mathbb{P}(y_i = \hat{y}(\mathbf{x}_i)) = 0$ be true for all $y_i$. However, to achieve the highest calibration error on a set of data points in this scenario, one can first isolate the misclassified data points and if the classifier is made to output confidence scores of 100% on all of them, using

the calibration attack for example, it would create a total calibration error of 100% on this set of misclassified data points. With regard to the correctly classified points, where accuracy is 100%, one can create the largest difference between the accuracy and average confidence by making the average confidence on this set as low as possible. Since confidence scores can only range from $1/K$ to 1, the largest possible difference between the average confidence score and the accuracy of 100% is $1 - 1/K$. Again, if $p_k = 1/K$, and every $\hat{p}_k(\mathbf{x}_i) = 1/K$, while $y_i = \hat{y}(\mathbf{x}_i) \forall \mathbf{x}_i$, then this makes the calibration error: $\mathbb{P}(y_i = \hat{y}(x_i) \mid \hat{p}_k(\mathbf{x}_i) = p_k) - p_k = 1 - 1/K$. It is not possible to create a higher level of calibration error since if $p_k > 1/K$ on some number of the correctly classified datapoints, then $\mathbb{P}(y_i = \hat{y}(\mathbf{x}_i) \mid \hat{p}_k(\mathbf{x}_i) = p_k)$ will still be 1, while $p_k > 1/K$ will lead to less calibration error on that subset of datapoints. With errors on both mutually exclusive subsets of data maximized, the theoretically highest miscalibration will be created on the full data.

To derive the upper bound of the ECE value that can be achieved by `MMA`, if we assume $q$ is the accuracy of a $K$-way classifier $\mathcal{F}$ on the dataset $\mathcal{D} = \{\langle \mathbf{x}_n, y_n \rangle\}_{n=1}^{N}$, then post successful attack all of the datapoints will fall into one of two bins representing average confidence scores of 1 and $1/K$, assuming the confidence range of each bin is 1%. The accuracy in these bins would be 0 and 1, respectively. And the proportion of data points falling into each respective bin is $1 - q$ and $q$. Based on the ECE formula the maximum error would be:

$$\begin{aligned}
ECE_{max} &= \frac{n_{bin_{(100/k)\%}}}{N}|acc(bin_{(100/k)\%}) - conf(bin_{(100/k)\%})| + \frac{n_{bin_{100\%}}}{N}|acc(bin_{100\%}) - conf(bin_{100\%})| \\
&= q * |1 - 1/K| + (1 - q) * |0 - 1| \\
&= 1 - q/K
\end{aligned}$$

(7)

$\square$

# D    Details of Experimental Setup

**Metrics.** To assess the degree of calibration error caused by each attack, we use two metrics, the popular binning-based Expected Calibration Error (ECE) (Pakdaman Naeini et al., 2015), and KS error (Gupta et al., 2021), which are formulated in detail in Section F.

**Datasets.** The datasets we use in our study are CIFAR-100, Caltech-101, and the German Traffic Sign Recognition Benchmark (GTSRB). CIFAR-100 and Caltech-101 are both popular image recognition benchmark datasets, containing various objects divided into 100 classes and 101 classes respectively. Given the importance of calibration in safety critical applications, we include a common use case of autonomous driving with the GTSRB dataset, which consists of images of traffic signs divided 43 classes. CIFAR-100 has 50,000 images for training, and 10,000 for testing. Caltech-101 totals around 9000 images. GTSRB is split into 39,209 training images and 12,630 test images

**Models.** ResNet-50 (He et al., 2016) is the primary model we train and test on due to it being a standard model for image classification. Non-convolutional attention-based networks have recently attained great results on image classification tasks, so we also experiment with the popular Vision Transformer (ViT) architecture (Dosovitskiy et al., 2021). Both of these models are the versions with weights pretrained on ImageNet (Deng et al., 2009). We use the VIT_B_16 variant of ViT, and the pretraining dataset used for each model is ImageNet_1K for ResNet and ImageNet_21K for ViT, and are fine-tuned on the target datasets. Pretrained models are advantageous to study given they can increase performance over training from randomly initialized weights and present a more practical use-case. The specific details behind our training procedures and our various model hyperparameters can be seen in Section E.

**Attack Settings.** Regarding the SA version of the attacks, for the $l_\infty$ and $l_2$ norm attacks we use the default SA settings for $\epsilon$ and $p$, which are $\epsilon = 0.05$ and $p = 0.05$ for $l_\infty$ and $\epsilon = 5.0$ and $p = 0.1$ for $l_2$. For our primary results we run the attacks on a representative 500 test cases from the test set of each dataset. Each attack is ran for 1000 iterations, far less than the default 10,000 in Andriushchenko et al. (2020), but since there is no need to change the label, less iterations are required, bolstering the use-case and threat for this form of attack.

The settings for the PGD version of the attacks differ due to the accommodations that need to be made to prevent the PGD algorithm from changing the label while still being able to have a large effect on the confidence. In terms of general settings, we again use $\epsilon = 0.05$ as the adversarial noise value for an $l_\infty$ norm. We use an $\alpha$ attack step size value of $5/255$. For our white-box results we use 10 iterations of the attack. In addition to these settings, some were made to the attack algorithm as simply preventing PGD from changing the class label while trying to calculate the adversarial noise often leads to poor performance in practice as many updates are prevented. Instead, a dropout factor is added to the $(h * w * c)$ adversarial noise matrix after each attack iteration that only applies a select portion of the updates, lessening the effect of updates that are too strong and have a high chance of flipping the label. The value for the dropout is dependent on whether it is the overconfidence or underconfidence attack. The most effective values in our experiments were found to be a dropout value of 0.95 for the underconfidence attack, and 0.2 for the overconfidence attack.

The results in our experiment section are obtained on three runs of each model with different random seeds.

## E  Specific Training Details

### E.1  General Settings

As mentioned previously, for our general attack implementation we use SA, which works by using a randomized search scheme to find localized square-shaped perturbation at random positions which are sampled in such a way as to be situated approximately at the boundary of the feasible set. We still use the original sampling distributions, however we remove the initialization (initial perturbation) for each attack since it is prone to changing the predicted labels. Naturally, we use the untargeted versions of the attacks, whereby the perturbations lead to increases in the probabilites of random non-predicted classes for the underconfidence attack, since we only care about the probability of the top predicted class.

The details of the training procedure for each of the models and datasets is as follows: For CIFAR-100 and GTSRB, we use the predefined training and test sets for both but use 10% of the training data for validation purposes. For Caltech-101, which comes without predetermined splits, we use an 80:10:10 train/validation/test split. For all of the datasets, we resize all images to be 224 by 224. We also normalize all of the data based on the ImageNet channel means and standard deviations. We apply basic data augmentation during training in the form of random cropping and random horizontal flips to improve model generalizability. The hyperparameters we used for training the ResNet-50 models include: a batch size of 128, with a CosineAnnealingLR scheduler, 0.9 momentum, 5e-4 weight decay, and a stochastic gradient descent (SGD) optimizer. For ViT, the settings are the same, except we also use gradient clipping with the max norm set to 1.0. We conduct basic grid search hyperparameter tuning over a few values for the learning rate (0.1,0.01,0.005,0.001) and training duration (in terms of epochs). Generally, we found that a learning rate of 0.01 worked best for both types of models. The training times vary for each dataset and model. For the ResNet-50 models we trained for 15 epochs on CIFAR-100, 10 epochs on Caltech-101, and 7 epochs on GTSRB. Likewise for ViT, we trained for 10 epochs on CIFAR-100, 15 epochs on Caltech-101, and 5 epochs on GTSRB. The results reported in Sections 3 and 5 are shown for models on the epoch at which they attained the best accuracy on the validation set. All of the training occurred on 24 GB Nvidia RTX-3090 and RTX Titan GPUs. Finally, we use 15 bins to calculate the ECE.

### E.2  Defence Training Settings

In this section, we describe each of the defences we used in Section 5, and the settings we use to train them (if applicable).

**Temperature Scaling (TS) (Guo et al., 2017).** TS is a post-process recalibration technique applied to the predictions of an already trained model that reduces the amount of high confidence predictions without affecting accuracy. TS works by re-scaling the logits after the final layer of the neural network to have a higher entropy by dividing them by a temperature parameter $T$, that is tuned by minimizing negative log likelihood (NLL) loss on the validation set. Temperature scaling only works well when the training and test

distributions are similar (Kumar et al., 2019), but by reducing overconfidence it may have an advantage against overconfidence attacks.

**Multi-domain Temperature Scaling (MD-TS) (Yu et al., 2022)** MD-TS is a method based on TS but is designed to be more robust in situations when data comes from multiple domains, as in this case with images corrupted using different types of calibration attacks. It modifies the original TS method by first finding the ideal temperature across each domain, then training a linear regression classifier using the feature embeddings of each datapoint and the corresponding ideal temperatures based on the respective domain, yielding a classifier that can dynamically calculate an ideal temperature for each instance at test time. We modify this domain for our task of defence by creating three different domains for the base images and their underconfidence attacked and overconfidence attacked counterparts. We select 500 validation instances and find the temperature for each domain and conduct the rest of the method as in its original incarnation. The feature embeddings are as before, using the penultimate layer outputs of model before the classification layer. We experiment with converting the feature embeddings to Fourier domain using the fast Fourier transform before feeding them to the classifier, similar to the principle behind the detection method in Harder et al. (2021) to make it easier to identify adversarially attacked datapoints, though we find that the conversion brings little benefit over the base variation of the method.

**Calibration of Neural Networks using Splines (Spline) (Gupta et al., 2021).** Spline is another post-process recalibration technique that uses a recalibration function to map existing neural network confidence scores to better calibrated versions by fitting a spline function that approximates the empirical cumulative distribution. It is lightweight, and often performs better than TS.

**Difference between confidence and accuracy (DCA) (Liang et al., 2020).** DCA is a training-based calibration method that adds an auxiliary loss term to the cross-entropy loss during training that penalizes any difference between the mean confidence and accuracy within a single batch, inducing a model to not produce confidence scores that are miscalibrated. We set the weight of DCA to 10 based on the recommendation by Liang et al. (2020). Training settings are kept the same as described in the general settings.

**Sharpness Aware Minimization (SAM) (Foret et al., 2021).** SAM is a technique that improves model generalizability by simultaneously minimizing loss value and loss sharpness. It finds parameters that lie in neighbourhoods having uniformly low loss by computing the regularized "sharpness-aware" gradient. The motivation behind using this technique as a defence is that models with parameters that lie in uniformly low loss areas may present additional difficulty in creating adversarial examples, and may be more regularized as a whole. We use a neighbourhood size $\rho = 0.05$. We kept the training settings the same as we described in the general settings.

**RobustBench (Croce et al., 2021).** To understand how state-of-the-art adversarial defences work against our attack, we take the top 3 performing (in terms of adversarial robustness) WideResNet (Zagoruyko & Komodakis, 2016) defences on the popular RobustBench defence model benchmark for CIFAR-100 under the $l_\infty$ $\epsilon = 8/255$ attack model. We only choose the WideResNet models given their closer similarity to the primary model we study in this work, ResNet-50. The defences we choose are those of Gowal et al. (2020) (ranked first), Rebuffi et al. (2021) (ranked third) and Pang et al. (2022) (ranked fifth). These defences use a combination of adversarial training and ensembling to produce models that are robust against a wide range of conventional adversarial attacks. In addition, they use different techniques, like combining larger models, using Swish/SiLU activations and model weight averaging, and data augmentation to significantly improve robust accuracy.

**Adversarial Training (AT).** AT is among the most common and effective defences against a wide range of adversarial attacks where models are trained on adversarially attacked images. We implement our version similar to Madry et al. (2018) and Xie et al. (2019), where we run PGD-based adversarial training, given how this form of defence has been shown to be effective across a wide range of attacks due to PGD being close to a universal first-order $l_\infty$ attack. We train exclusively on images attacked with an n-step $l_\infty$ PGD attack each batch, with the number of steps chosen depending on the model and dataset. Since we already test RobustBench models that often make use of AT with a large amount of steps, we specifically tune our AT models to have less steps to compromise less on accuracy and miscalibration. We wish to see whether

more lightly-tuned AT can still provide major benefits given calibration attacks are not as severe. For the PGD attack, we attack each image in a batch using an $\epsilon$ norm of 0.1. We use an attack stepsize relative to $\epsilon$ of 0.01 / 0.3, with random starts. The number of attack iterations ran for each batch was carefully chosen to balance performance and adversarial robustness. We used 15 iterations on all of the ResNet models, while for ViT we generally required much fewer, with three for the CIFAR-100 models, and five for the remaining two datasets. In terms of remaining training details, we keep them largely the same as described in the general settings, although the training durations were sometimes varied by a few epochs to optimize accuracy. We use the Foolbox implementation of the PGD attack (Rauber et al., 2020; 2017).

**Adversarial Attack Against Attacks (AAA) (Chen et al., 2022).** A recent adversarial defence specifically tuned towards black box score based methods like Square Attack, this is a post processing method that works on an already trained neural network's logits that uses a function that misleads the attack methods towards incorrect attack directions by slightly modifying the output logits. The method is shown to be very effective against score-based query methods at a low computational cost, and is purported to maintain good calibration, which makes it of particular interest in this case as a defence against calibration attacks.

**Calibration Attack Adversarial Training (`CAAT`)**. Our novel form of adversarial training that uses calibration attacks to generate adversarial examples rather than the regular attack algorithm. Although the general methodology is still the same as PGD-based adversarial training, the primary difference is that for each minibatch, both the underconfidence PGD calibration attack and its overconfidence version are applied to the images and the loss between the two sets of images is added. As this uses calibration attack, the labels of these images are unaffected. The settings we use for the attacks are the same as those described in D for the white-box version. Regarding the settings for each model and dataset, they are largely similar to those of regular AT, although the number of attack iterations is kept consistent at 10, even for ViT. The number of training epochs are the same as those we use for regular fine-tuning.

**Compression Scaling (`CS`)**. This is a novel post-process defence that does not require training and is specifically designed to maintain the regular confidence score distribution and thereby preventing extreme miscalibration while undergoing a calibration attack. Since calibration attacks does not flip the original label, for any given classifier, the strongest effect of calibration attacks will be reducing the confidence score on correctly classified "easy" datapoints while making the model more overconfident on difficult, misclassified datapoints. This creates a shift in the distribution where for a given high performing classifier the average confidence will drop dramatically while the accuracy remains high, and some misclassified datapoints will shift to a higher confidence level. In any case, a distribution that was originally skewed towards high confidence scores is now essentially shifted lower. Therein lies the goal of `CS`, to essentially shift back the distribution by scaling it such that it lies in high confidence space as before. If we assume that already low confidence correctly classified datapoints will be more affected by a calibration attack than one that is much higher confidence, and if we assume that incorrectly classified datapoints will have lower confidence then due to the relative inefficiency of the overconfidence attacks they will likely not reach extremely high confidence levels unless the attack is ran for a very large amount of iterations, then the relative ordering between many of the datapoints is still preserved even if the distribution is shifted, meaning the misclassified datapoints may still get mapped to the lower end of the confidence scale. The advantage of this method is that it largely does not incur a lot of calibration error even on clean data while being among the most effective and consistent defence methods against calibration attack. In addition, if one wants to do downstream decision making then one can still filter out the bottom $p$ percentage of images with a confidence score. For the number of bins, we mostly choose 3 or 4 as this leads to the smallest error post attack. We find the scaling factor by iterating through a large range of possible values so that the new desired confidence score for the datapoint is then achieved within the new confidence range.

**Binning Details.** In our two defence algorithms, the range of possible confidence scores are first split into equally sized bins. In our case, we divide confidence scores into 15 bins and chose the top 3 (or 4) highest confidence bins as the compressed bins as mentioned above.

## F    Calibration Metric Formulation

Here we formulate the two calibration metrics that we use in our experiments. As Equation 2 is an idealized representation of miscalibration that is intractable, approximations have been developed which are grouped into the more common binning-based metrics, and non-binning based metrics.

Expected calibration error (Pakdaman Naeini et al., 2015) is the most widely used calibration metric in research. It is a binning-based metric where confidence scores on the predicted classes are binned into $M$ number evenly spaced bins, which is a hyperparameter that must be carefully chosen. In each bin, the difference between the average confidence score and accuracy of all data points within the bin is calculated, representing the bin-wise calibration error. Afterwards, the weighted sum over the error in each bin constitutes the expectation of the calibration error of the model. The equation for ECE is as follows given $B_m$ are the data points in the $m^{th}$ bin, and $n_m$ is the number of data points in that bin.

$$ECE = \sum_{m=1}^{M} \frac{n_m}{N} |acc(B_m) - conf(B_m)|. \tag{8}$$

ECE can underestimate the levels of miscalibration due to being sensitive to the number of bins (Ovadia et al., 2019) and by having underconfident and overconfident data points overlapping in one bin (Nixon et al., 2019). Kolmogorov-Smirnov Calibration Error (Gupta et al., 2021) is an alternative evaluation metric, that instead of binning, leverages the Kolmogorov-Smirnov statistical test for comparing the equality of two distributions. The error is determined by taking the maximum difference between the cumulative probability distributions of the confidence scores and labels. Specifically, the first step is to sort the predictions according to the confidence score on class $k$, i.e., $\hat{p}_k$, leading to the error being defined as:

$$\begin{aligned} \text{KS} \ \ \text{error} &= \max_i |h_i - \tilde{h}_i|, \\ \text{where,} \ h_0 &= \tilde{h}_0 = 0, \\ h_i &= h_{i-1} + \mathbf{1}(y_i = k)/N, \\ \tilde{h}_i &= \tilde{h}_{i-1} + p_k(x_i)/N. \end{aligned} \tag{9}$$

## G    Adversarial Attack Detection Details

**Local Intrinsic Dimensionality (LID) (Ma et al., 2018).** This detection method exploits the estimated Local Intrinsic Dimensionality (LID) characteristics across different layers of a model of a set of adversarial examples, which are found to be notably different than that of clean datapoints or those with added random noise. First, a training set is made up of clean, noisy, and adversarial examples, and a simple classifier (logistic regression) is trained to discriminate between adversarial and non-adversarial examples. For each training minibatch, the input features to the classifier are generated based on the estimated LID across different layers for all of the datapoints. The hyperparameters for this method are batch size and the number of nearest neighbours involved in estimating the LID. We choose a consistent batch size of 100 in line with previous work such as (Harder et al., 2021), and for each case we test the possible nearest neighbors from the following list $\{10, 20, 30, 40, 50, 60, 70, 80, 90\}$ and report the results for the best value, which vary for different datasets and models. We use the implementation from Lee et al. (2018).

**Mahalanobis Distance (MD) (Lee et al., 2018).** The premise behind this method is to use a set of training datapoints to fit a class-conditional Gaussian distribution based on the empirical class means and empirical covariance of the training datapoints. Given a test datapoint, the Mahalanobis distance with respect to the closest class-conditional distribution is found and taken as the confidence score. A logistic regression detector is built from this which determines whether a datapoint is adversarial. The main hyperparameter for this method is the magnitude of the noise used, which we vary between $\{0.0, 0.01, 0.005, 0.002, 0.0014, 0.001, 0.0005\}$ for each case and pick the value that results in the highest detection accuracy. In addition, calculating the mean and covariance is necessary to use the method, which we utilize the respective training set to do for each dataset. We use the implementation of MD from (Lee et al., 2018).

**SpectralDefense (Harder et al., 2021).** This detection method makes use of Fourier spectrum analysis to discriminate between adversarial and clean images. The spectral features from Fourier coefficients, which are computed via two-dimensional discrete Fourier transformation applied to each feature map channel, are found for each image, and a detector based on logistic regression is trained using the Fourier coefficients. The magnitude Fourier spectrum based detector (InputMFS) is the version we use in our experiments.

# H    Additional Analysis and Results

In this section we provide additional results with white-box attacks, more details on the analyses described in Section 4.4, and qualitative analysis of the properties of our attacks, as well as a quantitative analysis under a common real world issue of imbalanced data distributions. Apart from the white-box results, the remaining analyses are conducted using our black-box setup.

## H.1    White-box Calibration Attack

The results for the white-box variation of our attacks can be found in Table 7 on the three datasets and across our two tested models, similar to how we presented our black-box results. For each scenario, we show the ECE, KS error and average confidence. We used 10 attack steps to generate the results for an $\epsilon$ noise value of 0.05.

Much like the SA results, the PGD attack manages to create significant miscalibration compared to before the attack with only a small number of attack steps. The results are less severe than for SA where the level of miscalibration achieved are worse despite the base PGD attack being far more effective at affecting classification accuracy. We believe this is because the modifications that are made to ensure that the calibration attack algorithm does not cause the predicted class to change greatly. This reduces the effectiveness of PGD as the most effective gradient updates that cause a great swing in the confidence score cannot be used since they are likely to change the predicted class, and instead much less significant updates that do not change the confidence a great deal serve as the primary noise that gets added to the adversarial images.

## H.2    Detailed Setup and Comparison of Efficiency of Underconfidence vs. Overconfidence Attacks.

To understand which form of attack is most query efficient when the amount of change in confidence is the same, for each attack type we identify all of images in the test set that are around a given confidence level. We use the corresponding attack to made the model produce either an increase of 10% in confidence, or a decrease of 10%. We choose two base confidence levels of 80% and 90% and find all the data points within 1% of each. When an attack causes a change at or past the set threshold for the given goal probability, the attack stops and the number of queries is recorded. The results can be seen in Table 3. The consistent pattern we observe for both base confidence levels is that it takes notably fewer queries to create underconfidence than overconfidence, and the former attack is more effective at affecting the average confidence.

Table 7: Results of white-box PGD variant of calibration attack.

| **ResNet** | | | |
|---|---|---|---|
| | **ECE** | **KS** | **Avg. Conf.** |
| *CIFAR-100* (Accuracy: 0.881±0.002) | | | |
| Pre-atk | 0.052±0.006 | 0.035±0.006 | 0.916±0.006 |
| UCA | 0.213±0.003 | 0.175±0.007 | 0.747±0.011 |
| OCA | 0.072±0.003 | 0.070±0.001 | 0.951±0.002 |
| MMA | 0.187±0.008 | 0.161±0.007 | 0.746±0.007 |
| RCA | 0.187±0.016 | 0.156±0.013 | 0.759±0.015 |
| *Caltech-101* (Accuracy: 0.966±0.004) | | | |
| Pre-atk | 0.035±0.002 | 0.031±0.004 | 0.936±0.001 |
| UCA | 0.388±0.019 | 0.380±0.022 | 0.599±0.022 |
| OCA | 0.018±0.003 | 0.019±0.002 | 0.984±0.001 |
| MMA | 0.375±0.022 | 0.376±0.022 | 0.591±0.021 |
| RCA | 0.353±0.019 | 0.352±0.021 | 0.619±0.022 |
| *GTSRB* (Accuracy: 0.972±0) | | | |
| Pre-atk | 0.019±0.006 | 0.008±0.002 | 0.98±0.002 |
| UCA | 0.233±0.020 | 0.232±0.016 | 0.752±0.014 |
| OCA | 0.020±0.002 | 0.019±0.003 | 0.991±0.003 |
| MMA | 0.226±0.006 | 0.227±0.007 | 0.750±0.008 |
| RCA | 0.217±0.014 | 0.218±0.009 | 0.763±0.008 |
| **ViT** | | | |
| *CIFAR-100* (Accuracy: 0.935±0.002) | | | |
| Pre-atk | 0.064±0.006 | 0.054±0.005 | 0.882±0.004 |
| UCA | 0.277±0.001 | 0.274±0.004 | 0.671±0.006 |
| OCA | 0.045±0.003 | 0.017±0.002 | 0.928±0.002 |
| MMA | 0.260±0.006 | 0.262±0.007 | 0.675±0.005 |
| RCA | 0.236±0.013 | 0.239±0.015 | 0.699±0.013 |
| *Caltech-101* (Accuracy: 0.961±0.024) | | | |
| Pre-atk | 0.137±0.059 | 0.136±0.06 | 0.825±0.083 |
| UCA | 0.489±0.071 | 0.489±0.071 | 0.472±0.095 |
| OCA | 0.086±0.045 | 0.082±0.049 | 0.879±0.073 |
| MMA | 0.488±0.070 | 0.488±0.070 | 0.473±0.094 |
| RCA | 0.435±0.048 | 0.435±0.048 | 0.527±0.071 |
| *GTSRB* (Accuracy: 0.947±0.006) | | | |
| Pre-atk | 0.040±0.005 | 0.026±0.017 | 0.922±0.024 |
| UCA | 0.321±0.047 | 0.315±0.045 | 0.641±0.052 |
| OCA | 0.037±0.013 | 0.020±0.011 | 0.936±0.024 |
| MMA | 0.302±0.037 | 0.302±0.036 | 0.645±0.043 |
| RCA | 0.292±0.030 | 0.293±0.029 | 0.657±0.037 |

### H.3 Detailed Analysis on Varying Epsilon and Attack Iterations

**Epsilon.** The $\epsilon$ parameter plays a major role in adversarial attacks, as it controls how much noise can be added when creating perturbations. Although setting a higher $\epsilon$ value for an attack lets it easier and more efficient for the algorithm to create adversarial examples, it potentially cause the visual changes to images more perceptible, so a small $\epsilon$ is preferable while still being able to produce good adversarial examples. In the case of calibration attack, there is no need to go as far as flipping a label, so lower $\epsilon$-bounds have the potential to create some miscalibration. To provide further details on our results in Figure 2, for the leftmost figure as mentioned previous we tested on CIFAR-100 using ResNet-50. The five different $\epsilon$ values we use are {0.005, 0.01, 0.05, 0.1, 0.25} after being attacked using all four of the attacks with the other settings the same as in Appendix D, with the results averaged over three models. In addition to the miscalibration being strong for most of the attacks at low $\epsilon$ values, we can see that `MMA` consistently outperforms the rest across the different values. `UCA` does not have much change with higher $\epsilon$, but it is largely because the models has already almost reached the peak level of attacking the confidence with low epsilon values, and as such does not have a large effect on ECE. As middle figure largely displays the same trends as ResNet, revealing that the results are not architecture dependant. The rightmost figure uses ResNet and goes over the same $\epsilon$ values as before, except `MMA` is run over both Caltech-101 and GTSRB models. Again the trends are similar, although the increase in ECE is not as severe as for CIFAR-100.

**Iterations.** Expanding on the results in the rightmost figure in Figure 2, the number of iterations of `MMA` is varied from 100, 500, 1000, to 5000, whilst attack both ViT and ResNet models trained and tested on CIFAR-100 with the same settings as in Appendix D. We note how the ECE begins to saturate at close to 500 iterations, after which the benefits of running the attack longer are minor, though even at 100 iterations the ResNet model becomes heavily miscalibrated despite the standard $\epsilon$ value of 0.05 being used. In our tests showing the effectiveness of the original SA versus its calibration attack version, seen in Figure 4, we specifically compare over accuracy and ECE between the maximum miscalibration attack and the regular untargeted SA across the four aforementioned iteration values for both ResNet and ViT on CIFAR-100. As expected, SA greatly reduces the accuracy even with a small number of iterations. Nevertheless, in terms of ECE, the calibration attacks consistently produce higher amounts of miscalibration compared to the original SA algorithm across the different iteration amounts.

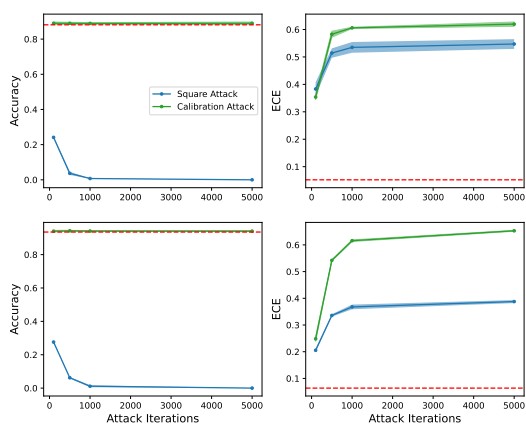

Figure 4: The contrast between the effects on accuracy and ECE between the original version of the Square Attack algorithm and the maximum variation of the calibration attack algorithm at 1000 attack iterations. (Top) ResNet-50 results. (Bottom) ViT results.

### H.4 t-SNE Visualizations

To help visualize the effect of each of the attack types in latent space and to confirm they are having the expected effects, we run a t-SNE analysis (van der Maaten & Hinton, 2008) on the representations of ResNet-50 right before the classification layer. The datasets we use throughout this study, with their large number of classes, are not ideal for visualization purposes. Instead, we create a binary subset using CIFAR-100 by taking all of the images from two arbitrary classes, bicycles and trains. We create a separate training set and test set to perform this procedure independently, and fine-tune a ResNet-50 model on the training set. The specific details are similar to those described in Section E for CIFAR-100 ResNet. We train the model for 5 epochs with a learning rate of 0.005. The attack settings are the same as in Section D for the $l_\infty$ version, and we only run the attacks for 500 iterations. We run the t-SNE analysis on a balanced slice of 200 images from the new test set for easy visualization purposes, before and after all four attack types. The model achieves 95% accuracy on the full test set. Figure 5 shows the graphs. It can be seen the effect on the representations

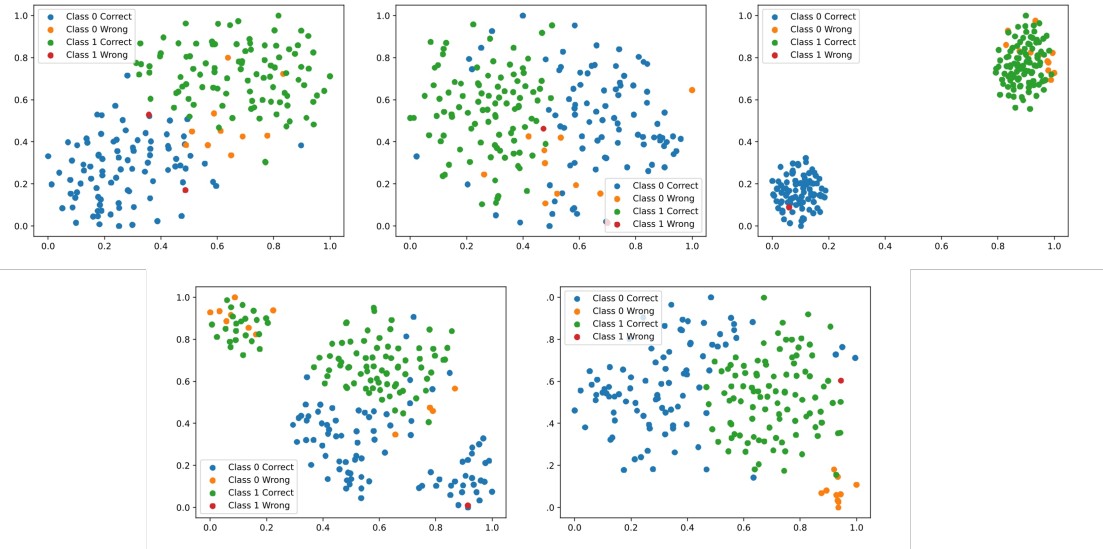

Figure 5: t-SNE visualization of the effect of different forms of calibration attacks on a ResNet model trained and tested on a binary subset from CIFAR-100, with the test set (consisting of 200 data points) results being displayed. In the order from top left to bottom right, the plots for the pre-attack (vanilla model), and the `UCA`, `OCA`, `RCA`, and `MMA` variations can be seen.

for the adversarially attacked data is as expected. `OCA` causes the representations for both class predictions, even incorrect ones, to be split apart as much as possible, while `UCA` causes a more jumbled representation between the two classes with most falling closely to the decision boundary. `MMA` has a similar effect to the underconfidence attack, except the misclassified images are pushed far away from the decision boundary to make it appear as if the model is more confident in its decisions. Lastly, `RCA` causes two distinct random clusters for each prediction type to form, as random data points are pushed to be more overconfident or more underconfident than they originally were. With these results, we can see visually confirm that the attacks possess their intended behaviour.

## H.5 Imbalance Ratio

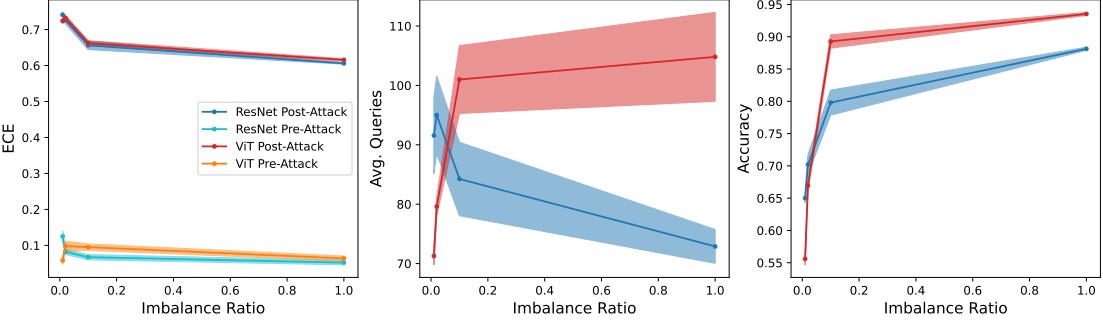

Figure 6: Graphs comparing the vulnerability of ResNet and ViT models trained with different imbalance ratios on CIFAR-100 to the maximum miscalibration attack at 1000 iterations and an $\epsilon$ of 0.05, and their corresponding overall trends in average queries and accuracy.

Dataset imbalance has a profound effect on how a model learns and how well it performs, with detrimental effects occurring when imbalance ratios are very high. With how common imbalanced data distributions are in real world scenarios, we believe it is worth studying the influence of imbalance ratio and its relationship with robustness against calibration attacks as an additional point of analysis. We choose CIFAR-100 as our

primary dataset for this analysis, and we follow the procedures in Tang et al. (2020) and Cao et al. (2019) to create training sets with long-tail imbalance. This is a form of imbalance where the datapoint sizes in the classes follow an exponential decay. We use the variable $\rho$ to denote the ratio between datapoint sizes of the class with the smallest datapoint size, and those of the one chosen to be the biggest. We create training sets with $\rho$ values of 0.01, 0.02, and 0.01 (for 1:100, 1:50 and 1:10 ratios of smallest to biggest class). We then train 3 ResNet-50 and 3 ViT models on each imbalanced set. The training details are again the same as those described in the general settings Section E, although the training times are different. 15 epochs is used to training the 1:100 ratio models, while 10 epochs is used for the rest. We subject the models to the maximum miscalibration attack using the same settings as in Section D for CIFAR-100 (test data is balanced), and calculate the resulting average and deviation of the pre and post attack ECE, average number of queries, and accuracy. The graphs displaying the results can be seen in Figure 6. Unsurprisingly, the higher the imbalance ratio, the lower the accuracy is on the balanced data. In terms of robustness, the more balanced the data the more resistant it is against getting miscalibrated from the attacks, for both the ResNet and ViT architectures. This is similar to the trends in the inherent miscalibration present before the attacks, although the calibration differences between the different ratio models are not as severe, and ViT at the 1:500 imbalance ratio is the best calibrated beforehand but becomes the worst after the attack. The trend in the number of queries it takes for a successful attack is reversed for ResNet and ViT, with ViT requiring more queries the more balanced the data is, while ResNet is vice-versa. Overall, dataset imbalance does not create favourable conditions for robustness, though the use of imbalance data techniques could potentially remedy some of these issues.

### H.6  GradCAM Visualization Details

Given the effectiveness of the attacks at leading a model to produce highly miscalibrated outputs, for both base styles of attacks, we endeavour to explore whether they also lead to any changes in where the model focuses on in an image when making its decision, and especially with `OCA`. Knowing this can lead to further insights as to how models are affected by calibration attacks. To accomplish this analysis, we use GradCAM (Selvaraju et al., 2017), a popular visualization method that produces a coarse localization map highlighting the most important regions in an image that the model uses when making its prediction by making use of the gradients from the final convolutional layer (or a specific layer of choice) of a network. We apply GradCAM to our ResNet-50 models fine-tuned on Caltech-101 to images from the Caltech-101 test set before and after the underconfidence and overconfidence attacks at the standard attack settings used in Section D and using the GradCAM implementation of Gildenblat & contributors (2021). Since the method calculates relative to a specific class, we do so in-terms of the original predicted class for each image. Figure 3 shows the results with some representative images. We specifically choose images where the attacks led to large change in predicted confidence (at least 10%). On the whole, we have observed that the coarse localization maps have minimal to no noticeable changes after the adversarial images are produced, especially in the case of the overconfidence attacked images. This leads us to believe the primary mechanism of the attacks changing the model confidence is in the final classification layer as opposed to the convolutional layers. This analysis also shows that it might be difficult to identify these attacks are occurring based on these types of gradient visualization methods.

### H.7  Certified Confidence Scores

In this section, we follow recent work in the domain of providing provable guarantees on the robustness of confidence scores by examining the effect on a smoothed classifier and its bounds when handling calibration attacked data, particularly in the case of `OCA`. Given the lack of label flipping, we expect that adversarial examples generated by calibration attack lie close to the original in the data manifold, thereby having a minimum effect on the certified confidence. To test this hypothesis, we closely follow Kumar et al. (2020) on certification to provide lower bounds on the confidence of a smoothed classifier. A strong lower bound can be produced by using the probability distribution of the confidence scores of a Gaussian cloud around the input image using Neyman-Pearson lemma to calculate this for a given certified radius. We select a subsample of 100 random datapoints from CIFAR-100 and use our base ResNet-50 models that are attacked using the underconfidence and overconfidence attacks using our standard settings, and average the expected

confidence score lower bound produced by the smoothed model across different Radii. We utilize the version of the method that uses CDF information and use a smoothing faction value of $\sigma = 0.25$, failure probability $\alpha = 0.001$ and 100,000 Monte-Carlo datapoints for the estimation. The results for the certified smoothed model can be seen in Figure 7. We observe that the average lower bound for the underconfidence attack is lower than the base version and the overconfidence attack, although it is not dramatically different due to the huge error bounds. Nevertheless, it appears that the smoothed model works well at being robust, particularly on the overconfidence attacked datapoints, but despite the expected confidence, the average certified accuracy for each set of data was 74%, 59%, and 76% respectively, so the smoothed classifier struggles at performing more accurately on the perturbed datapoints. Overall, this method of guarantee could be a useful tool for counteracting calibration attacks without using specific defences by determining how much confidence scores can reasonably be affected, particularly when using classifiers for safety-critical tasks that rely on confidence scores.

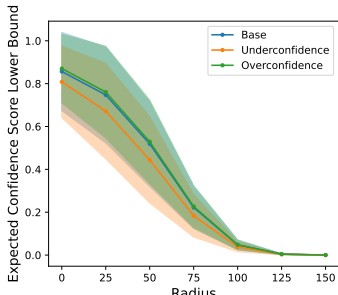

Figure 7: Expected Confidence Score lower bounds for a set of CIFAR-100 images (original and attacked using `UCA` and `OCA`) based on a ResNet-50 classifier.

## H.8 Results on Swin Transformers

We perform additional experiments using the Swin Transformer (Liu et al., 2021) architecture to further validate our attack results on additional architectures. Table 8 depicts the results on all types of calibration attack: `UCA`, `OCA`, `MMA`, and `RCA` on the Swin-Large variant of the model. The trends of the results on Swin Transformer agree with the existing results in Table 1. Specifically, `UCA`, `MMA` and `RCA` all bring both ECE and KS to high values. As observed previously, `OCA` successfully raises average confidences to high levels, but given the high accuracy of this victim model, increasing confidence will not have a drastic effect on calibration.

Table 8: Results of `UCA`, `OCA`, `MMA`, and `RCA` with the Swin-Large model on CIFAR-100. Accuracy of the victim model is included.

| | Avg qq | Med. qq | ECE | KS | Avg. Conf. |
|---|---|---|---|---|---|
| *Swin-Large (Acc: .943±.005)* | | | | | |
| Pre-atk | - | - | .026±.006 | .010±.003 | .945±.005 |
| UCA | 177.8±9.9 | 96.0±3.5 | .455±.006 | .430±.005 | .534±.008 |
| OCA | 34.4±4.9 | 1.0±0.0 | .039±.002 | .040±.001 | .980±.004 |
| MMA | 137.5±9.8 | 99.2±8.5 | .461±.007 | .434±.007 | .536±.007 |
| RCA | 138.8±14.4 | 102.2±5.3 | .433±.011 | .409±.006 | .557±.003 |

## H.9 Detailed Analyses on Recalibration and Defences

In this section, we provide more detailed analyses on the results of the different defences seen in Table 6.

The RobustBench models compromise substantially on accuracy, and have a high level of miscalibration on clean data. They do largely avoid getting extremely miscalibrated as a result of the attacks compared to the defenceless models, except the top model on the leaderboard. Nevertheless, their high inherent miscalibration means they are unfavourable in situations where the model must be well calibrated.

In terms of the calibration methods, TS tends to be among the best methods at reducing calibration error prior to the attacks, but after the attacks it offers very little benefit compared to the vanilla models. The MD-TS method is similar with its prior calibration being solid, but post-attack it only brings minor benefits

over TS in most cases. It appears that this is due in part to not finding the ideal temperature parameter for each image due to the difficulty of identifying the correct image domains, as in, recognizing which form of attack occurred. The Splines method is similar in its pre-attack calibration benefits to TS, but differs greatly in its performance post-attack. In some cases, like with CIFAR-100 using ResNet, it is easily the worst performing defence method. In other cases, particularly for Caltech-101 and GTSRB ViT, it is able to keep ECE at relatively reasonable values post-attack. This discrepancy shows that finding an ideal recalibration function has the potential to be a strong defence. The training-based DCA and SAM methods tend to bring few benefits after being attacked, even when they improve the calibration on clean data, the post-attack ECE and KS errors are not substantially different compared to the vanilla models.

The performance of the regular adversarial defence techniques is mixed. For robustness, AAA in most cases achieves the lowest post-attack ECE. Even in the best cases like Caltech-101 ResNet, ECE tends to be at least double compared pre-attack, and in most cases we still observed multiple-fold increases. This technique is also among the poorest calibrated on clean data. Regarding AT, our approach does not compromise on accuracy and miscalibration on clean data. It brings notable robustness, especially compared to the calibration methods, but it is not among the strongest.

Lastly, CS is the strongest methods at maintaining low calibration error on post-attack datapoints. Moreover, the technique tends to have better calibration error on clean data compared to AAA. It shows how it is key that high confidence values are retained to have decent calibration after the attacks. Altogether, despite some promising results with the defences, as a whole there are still limitations particularly with the strongest adversarial defences. The compromise of poor ECE on clean data for better calibration robustness against the attacks that we observe, as well as the general inconsistent performance means that further refinement on defences is warranted.

### H.10 Results of $l_2$ calibration attacks

Table 9: Results of the $l_2$-based calibration attacks on three datasets.

**ResNet**

|  | avg #q | median #q | ECE | KS | Avg. Conf |
|---|---|---|---|---|---|
| *CIFAR-100* | | | | | |
| **Accuracy:** 0.881±0.002 | | | | | |
| Pre-Attack | - | - | 0.052±0.006 | 0.035±0.006 | 0.916±0.006 |
| UCA | 182.7±13.0 | 94.0±11.0 | 0.399±0.012 | 0.356±0.010 | 0.566±0.008 |
| OCA | 44.6±3.3 | 1.0±0.0 | 0.129±0.007 | 0.129±0.007 | 0.995±0.000 |
| MMA | 137.3±5.8 | 92.7±10.6 | 0.496±0.001 | 0.391±0.002 | 0.604±0.003 |
| RCA | 125.2±7.3 | 99.7±4.9 | 0.431±0.016 | 0.350±0.012 | 0.614±0.011 |
| *Caltech-101* | | | | | |
| **Accuracy:** 0.966±0.004 | | | | | |
| Pre-Attack | - | - | 0.035±0.003 | 0.031±0.004 | 0.936±0.001 |
| UCA | 293.5±14.9 | 195.0±61.7 | 0.156±0.002 | 0.157±0.003 | 0.810±0.002 |
| OCA | 60.9±4.4 | 1.0±0.0 | 0.019±0.004 | 0.017±0.006 | 0.982±0.002 |
| MMA | 40.8±1.8 | 227.2±91.9 | 0.143±0.006 | 0.140±0.005 | 0.836±0.005 |
| RCA | 33.5±5.3 | 205.0±38.3 | 0.120±0.009 | 0.121±0.008 | 0.848±0.008 |
| *GTSRB* | | | | | |
| **Accuracy:** 0.972±0.000 | | | | | |
| Pre-Attack | - | - | 0.019±0.006 | 0.008±0.002 | 0.980±0.002 |
| UCA | 291.5±22.4 | 196.7±16.6 | 0.190±0.032 | 0.187±0.029 | 0.793±0.030 |
| OCA | 19.5±3.3 | 1.0±0.0 | 0.022±0.002 | 0.022±0.002 | 0.997±0.000 |
| MMA | 91.4±21.1 | 142.8±44.8 | 0.239±0.038 | 0.225±0.034 | 0.771±0.035 |
| RCA | 97.5±16.8 | 211.0±41.1 | 0.200±0.014 | 0.191±0.011 | 0.794±0.011 |

**ViT**

|  | avg #q | median #q | ECE | KS | Avg. Conf |
|---|---|---|---|---|---|
| *CIFAR-100* | | | | | |
| **Accuracy:** 0.935±0.002 | | | | | |
| Pre-Attack | - | - | 0.064±0.006 | 0.054±0.005 | 0.882±0.004 |
| UCA | 199.6±7.1 | 111.2±12.5 | 0.383±0.014 | 0.382±0.013 | 0.555±0.011 |
| OCA | 681.3±408.4 | 681.3±408.4 | 0.022±0.002 | 0.021±0.003 | 0.958±0.003 |
| MMA | 111.9±7.4 | 131.5±17.7 | 0.405±0.010 | 0.383±0.010 | 0.590±0.007 |
| RCA | 108.2±6.9 | 137.8±17.0 | 0.343±0.010 | 0.334±0.007 | 0.614±0.004 |
| *Caltech-101* | | | | | |
| **Accuracy:** 0.961±0.024 | | | | | |
| Pre-Attack | - | - | 0.137±0.059 | 0.136±0.060 | 0.825±0.083 |
| UCA | 258.7±47.2 | 207.8±64.0 | 0.233±0.057 | 0.233±0.057 | 0.729±0.081 |
| OCA | 23.2±15.0 | 1.0±0.0 | 0.100±0.048 | 0.100±0.048 | 0.859±0.073 |
| MMA | 31.5±2.2 | 236.5±52.0 | 0.224±0.058 | 0.224±0.058 | 0.740±0.080 |
| RCA | 21.9±8.1 | 293.8±24.5 | 0.196±0.038 | 0.196±0.038 | 0.764±0.064 |
| *GTSRB* | | | | | |
| **Accuracy:** 0.947±0.006 | | | | | |
| Pre-Attack | - | - | 0.040±0.005 | 0.026±0.017 | 0.922±0.024 |
| UCA | 258.3±27.8 | 169.7±31.5 | 0.261±0.012 | 0.262±0.011 | 0.686±0.016 |
| OCA | 70.2±31.3 | 1.0±0.0 | 0.030±0.005 | 0.024±0.012 | 0.968±0.016 |
| MMA | 99.6±10.9 | 210.5±33.0 | 0.274±0.037 | 0.257±0.038 | 0.718±0.044 |
| RCA | 94.7±11.2 | 213.8±59.9 | 0.245±0.020 | 0.241±0.016 | 0.714±0.007 |

### H.11 Full Attack Scale Results

In this section, we present the results of the four attack types across both black-box and white-box setups at various model scales of ResNet and ViT. Furthermore, we include four variants of Swin Transformers for each attack scenario. We included the Tiny (*swin-tiny-patch4-window7-224*), Small (*swin-small-patch4-window7-224*), Base (*swin-base-patch4-window7-224*), and Large (*swin-large-patch4-window7-224*) variants of Swin Transformers in our comparison trained with a similar setup used to obtain the results described in Appendix H.8.

Table 10: Attack effectiveness of `MMA` (black-box setup) across different model sizes of Swin Transformers on CIFAR-100.

| | Avg #q | Med. #q | ECE | KS | Avg.Conf. |
|---|---|---|---|---|---|
| *Swin-Small (Acc: .899±.008)* | | | | | |
| Pre | | | .036±.007 | .024±.008 | .923±.001 |
| Post | 120.3±5.7 | 72.5±4.1 | .511±.005 | .459±.007 | .494±.003 |
| *Swin-Base (Acc: .939±.009)* | | | | | |
| Pre | - | - | .034±.001 | .012±.005 | .940±.004 |
| Post | 153.4±14.2 | 101±12.3 | .464±.015 | .437±.011 | .527±.014 |
| *Swin-Large (Acc: .943±.005)* | | | | | |
| Pre | - | - | .026±.006 | .010±.003 | .945±.005 |
| Post | 137.5±9.8 | 99.2±8.5 | .461±.007 | .434±.007 | .536±.007 |

Table 11: Attack effectiveness of `UCA` (black-box setup) across different model sizes on CIFAR-100.

| | Avg #q | Med. #q | ECE | KS | Avg. Conf. |
|---|---|---|---|---|---|
| *ResNet-18 (Accuracy: .854±.004)* | | | | | |
| Pre | - | - | .037±.003 | .016±.005 | .870±.001 |
| Post | 60.5±1.5 | 35.3±1.5 | .531±.005 | .470±.006 | .446±.007 |
| *ResNet-50 (Accuracy: .881±.002)* | | | | | |
| Pre | - | - | .052±.006 | .035±.006 | .916±.006 |
| Post | 74.3±3.4 | 42.7±1.5 | .540±.005 | .479±.001 | .465±.005 |
| *ResNet-152 (Accuracy: .883±.002)* | | | | | |
| Pre | - | - | .051±.005 | .046±.004 | .929±.005 |
| Post | 90.4±5.5 | 48.0±1.0 | .538±.007 | .476±.004 | .473±.003 |
| *ViT-base (Accuracy: .935±.002)* | | | | | |
| Pre | - | - | .064±.006 | .054±.005 | .882±.004 |
| Post | 118.5±2.4 | 62.0±3.1 | .572±.007 | .553±.004 | .404±.003 |
| *ViT-large (Accuracy: .936±0.001)* | | | | | |
| Pre | - | - | .037±.009 | .022±.012 | .921±.021 |
| Post | 154.0±6.1 | 76.8±4.5 | .536±.007 | .514±.013 | .444±.019 |
| *Swin-Tiny (Accuracy: .883±.002)* | | | | | |
| Pre | - | - | .040±.008 | .031±.013 | .914±.011 |
| Post | 109.1±6.5 | 54.5±2.2 | .500±.012 | .442±.003 | .501±.006 |
| *Swin-Small (Accuracy: .899±.008)* | | | | | |
| Pre | - | - | .036±.007 | .024±.008 | .923±.001 |
| Post | 139.8±5.8 | 73.7±2.0 | .507±.001 | .456±.005 | .496±.002 |
| *Swin-Base (Accuracy: .939±.009)* | | | | | |
| Pre | - | - | .034±.001 | .012±.005 | .940±.004 |
| Post | 191.8±18.2 | 104.7±14.5 | .464±.012 | .438±.010 | .525±.015 |
| *Swin-Large (Accuracy: .943±.005)* | | | | | |
| Pre | - | - | .026±.006 | .010±.003 | .945±.005 |
| Post | 177.8±9.9 | 96±3.5 | .455±.006 | .430±.005 | .534±.008 |

Table 12: Attack effectiveness of `RCA` (black-box setup) across different model sizes on CIFAR-100.

| | Avg #q | Med. #q | ECE | KS | Avg. Conf. |
|---|---|---|---|---|---|
| ResNet-18 (Accuracy: .854±.004) | | | | | |
| Pre | - | - | .037±.003 | .016±.005 | .870±.001 |
| Post | 56.3±1.9 | 40.3±1.5 | .479±.015 | .420±.007 | .495±.006 |
| ResNet-50 (Accuracy: .881±.002) | | | | | |
| Pre | - | - | .052±.006 | .035±.006 | .916±.006 |
| Post | 68.9±4.6 | 42.7±1.2 | .558±.011 | .461±.003 | .514±.003 |
| ResNet-152 (Accuracy: .883±.002) | | | | | |
| Pre | - | - | .051±.005 | .046±.004 | .929±.005 |
| Post | 81.0±1.5 | 48.0±2.0 | .512±.003 | .451±.002 | .496±.003 |
| ViT-base (Accuracy: .935±.002) | | | | | |
| Pre | - | - | .064±.006 | .054±.005 | .882±.004 |
| Post | 106.4±3.0 | 70.3±1.5 | .549±.002 | .505±.003 | .471±.007 |
| ViT-large (Accuracy: .936±.001) | | | | | |
| Pre | - | - | .037±.009 | .022±.012 | .921±.021 |
| Post | 122.5±11.6 | 83.5±5.1 | .502±.005 | .479±.001 | .479±.008 |
| Swin-Tiny (Accuracy: .883±.002) | | | | | |
| Pre | - | - | .040±.008 | .031±.013 | .914±.011 |
| Post | 101.4±.7 | 62.7±2.9 | .474±.018 | .417±.009 | .526±.008 |
| Swin-Small (Accuracy: .899±.008) | | | | | |
| Pre | | | .036±.007 | .024±.008 | .923±.001 |
| Post | 124.6±5.4 | 79.2±3.4 | .483±.008 | .432±.009 | .519±.006 |
| Swin-Base (Accuracy: .939±.009) | | | | | |
| Pre | - | - | .034±.001 | .012±.005 | .940±.004 |
| Post | 161.6±25.5 | 116.6±21.3 | .440±.011 | .415±.009 | .548±.010 |
| Swin-Large (Accuracy: .943±.005) | | | | | |
| Pre | - | - | .026±.006 | .010±.003 | .945±.005 |
| Post | 138.8±14.4 | 102.2±5.3 | .433±.011 | .409±.006 | .557±.003 |

Table 13: Attack effectiveness of `OCA` (black-box setup) across different model sizes on CIFAR-100.

| | Avg #q | Med. #q | ECE | KS | Avg. Conf. |
|---|---|---|---|---|---|
| ResNet-18 (Accuracy: .854±.004) | | | | | |
| Pre | - | - | .037±.003 | .016±.005 | .870±.001 |
| Post | 21.6±1.4 | 1.0±.0 | .082±.003 | .083±.003 | .936±.003 |
| ResNet-50 (Accuracy: .881±.002) | | | | | |
| Pre | - | - | .052±.006 | .035±.006 | .916±.006 |
| Post | 16.0±.8 | 1.0±.0 | .124±.002 | .124±.002 | .996±.000 |
| ResNet-152 (Accuracy: .883±.002) | | | | | |
| Pre | - | - | .051±.005 | .046±.004 | .929±.005 |
| Post | 10.3±2.6 | 1.0±.0 | .077±.001 | .077±.001 | .961±.002 |
| ViT-base (Accuracy: .935±.002) | | | | | |
| Pre | - | - | .064±.006 | .054±.005 | .882±.004 |
| Post | 524.7±88.7 | 510.5±114.3 | .043±.007 | .043±.006 | .974±.001 |
| ViT-large (Accuracy: .936±.001) | | | | | |
| Pre | - | - | .037±.009 | .022±.012 | .921±.021 |
| Post | 345.0±96.1 | 295.75±117.7 | .039±.006 | .039±.004 | .962±.008 |
| Swin-Tiny (Accuracy: .883±.002) | | | | | |
| Pre | - | - | .040±.008 | .031±.013 | .914±.011 |
| Post | 27.3±9.8 | 1.0±.0 | .074±.004 | .075±.004 | .958±.003 |
| Swin-Small (Accuracy: .899±.008) | | | | | |
| Pre | | | .036±.007 | .024±.008 | .923±.001 |
| Post | 35.2±7.3 | 1.0±.0 | .065±.005 | .065±.005 | .965±.002 |
| Swin-Base (Accuracy: .939±.009) | | | | | |
| Pre | - | - | .034±.001 | .012±.005 | .940±.004 |
| Post | 34.6±4.3 | 1.0±.0 | .039±.004 | .039±.005 | .978±.005 |
| Swin-Large (Accuracy: .943±.005) | | | | | |
| Pre | - | - | .026±.006 | .010±.003 | .945±.005 |
| Post | 34.4±4.9 | 1.0±.0 | .039±.002 | .040±.001 | .980±.004 |

Table 14: Attack effectiveness of `MMA` (white-box setup) on different model sizes on CIFAR-100.

|  |  | ECE | KS | Avg. Conf. |
|---|---|---|---|---|
| ResNet-18 | Pre | .037±.003 | .016±.005 | .870±.001 |
|  | Post | .158±.006 | .144±.004 | .726±.005 |
| ResNet-50 | Pre | .052±.006 | .035±.006 | .916±.006 |
|  | Post | .187±.008 | .161±.007 | .746±.007 |
| ResNet-152 | Pre | .051±.005 | .046±.004 | .929±.005 |
|  | Post | .157±.011 | .137±.007 | .772±.002 |
| ViT-base | Pre | .064±.006 | .054±.005 | .882±.004 |
|  | Post | .260±.006 | .262±.007 | .675±.005 |
| ViT-large | Pre | .037±.009 | .022±.012 | .921±.021 |
|  | Post | .279±.054 | .277±.056 | .662±.056 |
| Swin-Tiny | Pre | .040±.008 | .031±.013 | .914±.011 |
|  | Post | .078±.005 | .059±.015 | .842±.018 |
| Swin-Small | Pre | .036±.007 | .024±.008 | .923±.001 |
|  | Post | .117±.005 | .101±.008 | .817±.005 |
| Swin-Base | Pre | .034±.001 | .012±.005 | .940±.004 |
|  | Post | .160±.035 | .154±.032 | .795±.024 |
| Swin-Large | Pre | .026±.006 | .010±.003 | .945±.005 |
|  | Post | .135±.009 | .133±.012 | .827±.013 |

Table 15: Attack effectiveness of `UCA` (white-box setup) on different model sizes on CIFAR-100.

|  |  | ECE | KS | Avg. Conf. |
|---|---|---|---|---|
| ResNet-18 | Pre | .037±.003 | .016±.005 | .870±.001 |
|  | Post | .232±.012 | .176±.006 | .740±.003 |
| ResNet-50 | Pre | .052±.006 | .035±.006 | .916±.006 |
|  | Post | .213±.003 | .175±.007 | .747±.011 |
| ResNet-152 | Pre | .051±.005 | .046±.004 | .929±.005 |
|  | Post | .210±.012 | .158±.003 | .781±.007 |
| ViT-base | Pre | .064±.006 | .054±.005 | .882±.004 |
|  | Post | .277±.001 | .274±.004 | .671±.006 |
| ViT-large | Pre | .037±.009 | .022±.012 | .921±.021 |
|  | Post | .299±.058 | .285±.063 | .668±.069 |
| Swin-Tiny | Pre | .040±.008 | .031±.013 | .914±.011 |
|  | Post | .142±.023 | .088±.021 | .854±.017 |
| Swin-Small | Pre | .036±.007 | .024±.008 | .923±.001 |
|  | Post | .179±.004 | .128±.008 | .829±.005 |
| Swin-Base | Pre | .034±.001 | .012±.005 | .940±.004 |
|  | Post | .217±.019 | .175±.023 | .812±.020 |
| Swin-Large | Pre | .026±.006 | .010±.003 | .945±.005 |
|  | Post | .191±.015 | .158±.01 | .816±.017 |

Table 16: Attack effectiveness of `RCA` (white-box setup) on different model sizes on CIFAR-100.

|            |      | ECE | KS | Avg. Conf. |
|------------|------|-----|-----|-----------|
| ResNet-18  | Pre  | .037±.003 | .016±.005 | .870±.001 |
|            | Post | .179±.016 | .138±.006 | .762±.003 |
| ResNet-50  | Pre  | .052±.006 | .035±.006 | .916±.006 |
|            | Post | .187±.016 | .156±.013 | .759±.015 |
| ResNet-152 | Pre  | .051±.005 | .046±.004 | .929±.005 |
|            | Post | .188±.004 | .144±.008 | .789±.016 |
| ViT-base   | Pre  | .064±.006 | .054±.005 | .882±.004 |
|            | Post | .236±.013 | .239±.015 | .699±.013 |
| ViT-large  | Pre  | .037±.009 | .022±.012 | .921±.021 |
|            | Post | .265±.047 | .259±.051 | .686±.057 |
| Swin-Tiny  | Pre  | .040±.008 | .031±.013 | .914±.011 |
|            | Post | .103±.0140 | .067±.018 | .855±.018 |
| Swin-Small | Pre  | .036±.007 | .024±.008 | .923±.001 |
|            | Post | .148±.014 | .112±.015 | .832±.010 |
| Swin-Base  | Pre  | .034±.001 | .012±.005 | .940±.004 |
|            | Post | .175±.022 | .152±.019 | .816±.013 |
| Swin-Large | Pre  | .026±.006 | .010±.003 | .945±.005 |
|            | Post | .159±.014 | .136±.014 | .842±.020 |

Table 17: Attack effectiveness of `OCA` (white-box setup) on different model sizes on CIFAR-100.

|            |      | ECE | KS | Avg. Conf. |
|------------|------|-----|-----|-----------|
| ResNet-18  | Pre  | .037±.003 | .016±.005 | .870±.001 |
|            | Post | .079±.006 | .079±.006 | .933±.002 |
| ResNet-50  | Pre  | .052±.006 | .035±.006 | .916±.006 |
|            | Post | .072±.003 | .070±.001 | .951±.002 |
| ResNet-152 | Pre  | .051±.005 | .046±.004 | .929±.005 |
|            | Post | .075±.001 | .075±.002 | .959±.001 |
| ViT-base   | Pre  | .064±.006 | .054±.005 | .882±.004 |
|            | Post | .045±.003 | .017±.002 | .928±.002 |
| ViT-large  | Pre  | .037±.009 | .022±.012 | .921±.021 |
|            | Post | .039±.011 | .021±.008 | .928±.019 |
| Swin-Tiny  | Pre  | .040±.008 | .031±.013 | .914±.011 |
|            | Post | .070±.004 | .066±.005 | .949±.003 |
| Swin-Small | Pre  | .036±.007 | .024±.008 | .923±.001 |
|            | Post | .057±.008 | .056±.008 | .955±.001 |
| Swin-Base  | Pre  | .034±.001 | .012±.005 | .940±.004 |
|            | Post | .037±.005 | .036±.003 | .974±.006 |
| Swin-Large | Pre  | .026±.006 | .010±.003 | .945±.005 |
|            | Post | .038±.003 | .037±.002 | .978±.004 |

