# OpenReview forum: "Calibration Attacks: A Comprehensive Study of Adversarial Attacks on Model Confidence"
_TMLR — Accepted by TMLR_

### Review · Reviewer_9Y8b · 2024-06-14

**Summary Of Contributions:**

This paper mainly studies the calibration attack, a specific type of adversarial attack that tries to make the target model miscalibrated rather than to induce misclassifications. Specifically, modern machine learning applications exploit the model's confidence to improve its trustworthiness, and therefore, the miscalibrated confidence will undermine the reliability of real-world machine learning applications.

The authors start with formalizing the objective of calibration attacks. In a calibration attack, the attack does not want to change the prediction but wants to make the target model less calibrated for adversarial examples. Here, the definition of class-wise calibration follows the conventional definition that a well-calibrated model will match the model's confidence to the probability that the prediction is correct. Therefore, the attacker will aim to make the model’s confidence on the adversarial example far from such probability. The authors propose four different objective functions that can yield miscalibration: *Underconfidence Attack* (UCA), *Overconfidence* Attack (OCA), *Maximum Miscalibration Attack* (MMA), and *Random Confidence Attack* (RCA). Also, the author proposes two defenses against calibration attacks: *Calibration Attack Adversarial Training* (CAAT) and *Compression Scaling* (CS).

The authors demonstrate the properties of calibration attacks, e.g., attack performance, sneakiness against detection, qualitative analysis, etc., with a set of experiments. Their simple implementation is effective enough to produce miscalibrated predictions of the target model. Also, the existing adversarial example detection algorithms do not work well against calibration attacks. The authors also experiment with possible defenses against calibration attacks, even though the proposed defense methods show limited performances.

**Audience:**

Yes

**Broader Impact Concerns:**

I don’t see a particular broader impact concern regarding this paper.

**Claims And Evidence:**

Yes

**Requested Changes:**

1. Please supplement Section 3.4 with more detailed real-world scenarios and emphasis on the threat.
2. If possible, do some additional experiments that show the transferability of miscalibration across different model architectures. For example, does UCA output miscalibrated inputs that show underconfidence across many different architectures?
3. Writing improvements
   * The authors should provide more details about the definition of metrics. This will make the result more understandable based on what each metric means.
   * In Tables 1, 2, and 3, highlight the key results with boldface (as the authors did in Table 4). This helps the readers to focus on the important part more quickly.
   * In Figure 2, use subfigure captions rather than referring them to subfigure 1, 2, etc.
   * Section 5 also contains experimental results. Why is Section 5 separated from Section 4? If this is for the distinction between attack-related experiments and defense-related experiments, you may change the title of Section 4. In my opinion, Section 5 does not have enough content to be an independent section, so I’d suggest merging Section 5 into the subsections of Section 4.

**Strengths And Weaknesses:**

### Strengths
1. The authors make novel contributions to calibration attacks. In particular, they properly formulate four different types of calibration attacks. They also invented two defense methods against calibration attacks, and this contribution to defense should be recognized.
2. The paper contains ample experiments that answer various questions regarding the properties of calibration attacks. First, different models, datasets, and metrics combinations are considered for each attack type. Also, the paper asks many research questions and performs experiments to demonstrate the answer.

### Weaknesses
1. As far as I know, this is the first paper that tries to bring the calibration attack to formal research. However, I don’t think that the paper justifies enough about the importance of the problem. I’d suggest adding more details in Section 3.4 with more illustrative examples. For example, provide more specific details about how real-world authority mechanisms work with well-calibrated inputs to show why miscalibration can be an existing real-world threat.
2. The miscalibration's transferability (across different model architectures) should be experimentally verified. If it does not show enough transferability, using multiple models simultaneously could be an easy defense against calibration attacks.
3. Though the writing quality is generally good, the paper writing can be improved further. Please see **Requested Changes** for more details.

---

### Review · Reviewer_nBsh · 2024-07-14

**Summary Of Contributions:**

This study investigates calibration attacks, which manipulate model confidence levels while preserving predicted labels. Four types of attacks are identified and evaluated across different model architectures. The findings underscore the need for heightened awareness and continued research efforts to enhance the resilience of models against such nuanced adversarial threats.

**Audience:**

Yes

**Claims And Evidence:**

Yes

**Requested Changes:**

See above weaknesses.

**Strengths And Weaknesses:**

Strengths:

- This paper is well-structured and easy to comprehend.
- The authors comprehensively explore four types of calibration attacks.

Weaknesses:
- The impact of calibration attacks on subsequent decision-making processes is not clearly evaluated.
- Additional evaluations across different model architectures, such as Swin Transformer, are necessary.
- The influence of varying model sizes on attack effectiveness is not investigated.
- Further exploration across different tasks, particularly within the realm of natural language processing (NLP) tasks and models, is needed

---

> ### Author Response · Authors · 2024-08-13
> **Response to Review nBsh (1/5)**
>
> We thank the reviewer for the comments.
>
> **Reviewer’s Comment**: Additional evaluations on Swin Transformer.
>
> **Response**:
>
> We followed the reviewer’s suggestion and performed additional experiments on the Swin Transformers. We will include the results in our revision. Table 7 below depicts the results on all types of calibration attack: UCA, OCA, MMA, and RCA. Note that we have also conducted more experiments on four different sizes of Swin Transformers (tiny, small, base, and large) in our response to the next question, together with other models, which we will also include into our revision. We respectfully argue that the experiments presented in our paper are comprehensive (the paper will have ~30 pages including appendices), spanning over the most used architectures and datasets.
>
> Table 7: Results of underconfidence (UCA), overconfidence (OCA), maximum miscalibration (MMA), and random confidence attack (RCA) with the Swin Large model on CIFAR-100. Accuracy of the victim model is included.
>
> ||Avg \#q|Med. \#q|ECE|KS|Avg. Conf.|
> |:--------------------------------:|:----------:|:----------:|:----------:|:----------:|:------------:|
> |Swin-Large (Acc: .943±.005)|||||||
> |Pre-atk|-|-|.026±.006|.010±.003|.945±.005|
> |UCA|177.8±9.9|96.0±3.5|.455±.006|.430±.005|.534±.008|
> |OCA|34.4±4.9|1.0±0.0|.039±.002|.040±.001|.980±.004|
> |MMA|137.5±9.8|99.2±8.5|.461±.007|.434±.007|.536±.007|
> |RCA|138.8±14.4|102.2±5.3|.433±.011|.409±.006|.557±.003|
>
> The trends of the results on Swin Transformer agree with the existing results in the paper (Table 1). Specifically, UCA, MMA and RCA bring both ECE and KS to high values. Same as in Table 1 in the paper, OCA successfully raises average confidences (the last column in the table) to high levels, but given the high accuracy of the victim models, increasing confidence will not have a drastic effect on calibration. (OCA will be more harmful on less accurate models.)
>
> **Reviewer’s Comment**: Influence of model sizes on attack effectiveness
>
> **Response**:
>
> We thank the reviewer for the comments. To help understand the influence of varying model sizes, we performed a comprehensive study on models of different sizes over three architectures across four attack variants.
>
> Below we show the results of the strongest variant of calibration attack (i.e., MMA). More detailed results on other model-vs-size combinations are included in the following response textbox.
>
> We will include Table 8 in the main paper and Table 9-15 in the appendix.
>
> Specifically, for ResNet, we compare ResNet-18, ResNet-50, and ResNet-152. For ViT, we included base (vit-base-patch16-224-in21k) and large models (vit-large-patch16-224-in21k). For Swin Transformers, we included tiny (swin-tiny-patch4-window7-224), small (swin-small-patch4-window7-224), base (swin-base-patch4-window7-224), and large (swin-large-patch4-window7-224) models.
>
> The training closely follows the settings detailed in Appendix D.1. We fine-tune three models for each variant and average the results. We ran attacks with the same settings used to produce Table 1 and 5 in the paper.
>
> Table 8: Attack effectiveness of MMA for the black-box attacks on different model sizes on CIFAR-100.
>
> |||**Avg\#q**|**Med.\#q**|**ECE**|**KS**|**Avg.Conf.**|
> |:-------------------------------:|:--:|:----------:|:----------:|:----------:|:----------:|:------------:|
> |ResNet18 (Acc: .854±.004)|||||||
> ||Pre|-|-|.037±.003|.016±.005|.870±.001|
> ||Post|52.9±1.4|36.0±0.0|.531±.009|.471±.004|.446±.003|
> |Resnet50 (Acc: .881±.002)|||||||
> ||Pre|-|-|.052±.006|.035±.006|.916±.006|
> ||Post|72.9±2.8|41.5±2.8|.606±.002|.497±.002|.502±.002|
> |ResNet152 (Acc: .883±.002)|||||||
> ||Pre|-|-|.051±.005|.046±.004|.929±.005|
> ||Post|80.5±3.8|47.0±3.5|.539±.002|.475±.002|.472±.008|
> |ViT-base (Acc: .935±.002)|||||||
> ||Pre|-|-|.064±.006|.054±.005|.882±.004|
> ||Post|104.8±7.5|62.7±4.7|.616±.003|.564±.000|.431±.001|
> |ViT-large (Acc: .936±0.001)|||||||
> ||Pre|-|-|.037±.009|.022±.012|.921±.021|
> ||Post|122.1±2.0|76.3±3.2|.531±.012|.508±.020|.450±.024|
> |Swin-Tiny (Acc: .883±.002)|||||||
> ||Pre|-|-|.040±.008|.031±.013|.914±.011|
> ||Post|97.6±1.4|58.7±1.2|.502±.012|.445±.006|.499±.010|
> |Swin-Small (Acc: .899±.008)|||||||
> ||Pre|||.036±.007|.024±.008|.923±.001|
> ||Post|120.3±5.7|72.5±4.1|.511±.005|.459±.007|.494±.003|
> |Swin-Base (Acc: .939±.009)|||||||
> ||Pre|-|-|.034±.001|.012±.005|.940±.004|
> ||Post|153.4±14.2|101±12.3|.464±.015|.437±.011|.527±.014|
> |Swin-Large (Acc: .943±.005)|||||||
> ||Pre|-|-|.026±.006|.010±.003|.945±.005|
> ||Post|137.5±9.8|99.2±8.5|.461±.007|.434±.007|.536±.007|
>
> Table 8 shows that the proposed calibration attacks are effective among different model sizes. The model sizes do not have a major impact on the susceptibility to calibration attacks, as the overall ECE and KS scores do not follow a clear pattern along with varying model sizes. However, when models scale up, the numbers of queries increase in general, suggesting that attacking complex models requires more rounds of attacks in most cases.

---

> ### Author Response · Authors · 2024-08-13
> **Response to Review nBsh (2/5)**
>
> **Reviewer’s Comment**: Impact of attacks on subsequent processes.
>
> **Response**:
>
> The typical and the most cited previous works in calibration and adversarial attacks often, if not all, focus on intrinsic evaluation metrics rather than extrinsic down-stream decision-making processes. In the case of calibration, typical assessments are miscalibration scores such as ECE and KS (Guo et al, 2017; Kumar et al, 2018; Minderer et al, 2021), and in adversarial attacks the evaluation is on the metrics with attack success rate (Andriushchenko et al, 2020; Madry et al, 2018; Chen et al, 2017).
>
> We performed our evaluation by following these conventions. In addition, the existing literature has significant discussions on how miscalibration affects downstream applications and human authority mechanism. Since our attack highly skews the confidence scores, the downstream tasks will be inherently in jeopardy. As a standalone paper (with ~30 pages after revision, including appendices), we respectfully argue that our paper follows the convention and has made adequate contributions.
>
> We will enhance the paper (as detailed below) by adding the discussions on the effect of skewed confidences on downstream tasks and human decision:
>
> --------------------------
> Previous studies (Penso et al, 2024; Jiang et al, 2012; Kompa et al, 2021) have shown the effect of miscalibration on down-stream authority mechanisms. Penso et al. (2024) discuss how medical imaging, if skipping expert review for the confident but incorrect model predictions, can have disastrous consequences, stressing the importance of being well-calibrated on incorrect predictions. Kompa et al. (2021) advocate for medical AI models being able to abstain from providing a diagnosis when there is a significant uncertainty for a patient, and to defer to additional human expertise to make a more informed diagnosis. This relies on a model that can give well-calibrated estimates.
>
> Moreover, prior works have examined the role of skewed confidence affecting the level of trust humans have in a system, which influences applications where humans and AI decision-making occurs symbiotically (Zhang et al, 2020; Rechkemmer et al, 2022). Antifakos et al. (2005), through a study on context-aware mobile phones, show how providing explicit confidence improves user trust in a system. Zhang et al. (2020) demonstrate when a model produces high confidence scores, users are more prone to rely on the AI for decision-making. These factors show why miscalibrated outputs can have great influence on downstream processing and authority mechanisms.
>
> As more specifical examples in the healthcare domain, chest abnormality detection (Dyer et al, 2021) tag radiographs with high-confidence normality to help radiologists make decisions. Call workers use confidence, along with other information, to make their decisions. The study discusses how the tool could be “glitching” out by providing incorrect confidence, compromising operability. In child maltreatment hotlines, screening models (De-Arteaga et al, 2020) are deployed where confidence is provided to suggest the likelihood that a child on the referral will experience adverse welfare-related outcomes. In the legal domain, LegalLens (Bernsohn et al, 2024) detects legal violations. Entities with high confidence are further processed with downstream models. In enterprise security, anomalous behaviour detectors (Shashanka et al, 2016) use confidence to send alerts about likely malicious activity.
>
> ----------------
> M. Andriushchenko, et al. Square Attack: a query-efficient black-box adversarial attack. ECCV 2020.
>
> P. Chen, et al. ZOO: Zeroth Order Optimization Based Black-Box Attacks to Deep Neural Networks without Training Substitute Models. AISEC. 2017.
>
> C. Guo, et al. On Calibration of Modern Neural Networks. ICML, 2017
>
> X. Jiang, et al. Calibrating predictive model estimates to support personalized medicine. JAMIA, 19:263–74, 2012.
>
> B. Kompa, et al. Second opinion needed: communicating uncertainty in medical machine learning. Digital Medicine, 4, 2021.
>
> C. Penso, et al. Confidence calibration of a medical imaging classification system that is robust to label noise. IEEE Tran. Med. Imag., 43:2050–60, 2024.
>
> A. Kumar, et al. Trainable Calibration Measures for Neural Networks from Kernel Mean Embeddings. ICML, 2018.
>
> A. Madry, et al. Towards Deep Learning Models Resistant to Adversarial Attacks. ICLR, 2018.
>
> M. Minderer, et al. Revisiting the Calibration of Modern Neural Networks. ArXiv, abs/2106.07998, 2021.
>
> A. Rechkemmer and M. Yin. When confidence meets accuracy: Exploring the effects of multiple performance indicators on trust in machine learning models. CHI, 2022.
>
> C. Xie, et al. Improving transferability of adversarial examples with input diversity. CVPR, 2019.

---

> ### Author Response · Authors · 2024-08-13
> **Response to Review nBsh (3/5)**
>
> **Reviewer’s Comment**: Further exploration across different tasks, e.g., natural language processing (NLP) tasks.
>
> **Response**:
>
> Previous works related to uncertainty/confidence in an adversarial context (Galil & El-Yaniv, 2021, Kumar et al. 2020, and Stutz et al., 2020) have often evaluated on images to make adequate contribution (and were extended to other modalities in separate papers). We respectfully argue that our paper makes adequate contributions as a stand-along paper, although extending it to text is very interesting, which we will discuss below in details. The image domain represents a scenario for demonstrating the destructive nature of the attacks where the inputs and perturbations are continuous, as such the confidence is manipulated by making imperceptible changes to the input leading to severe miscalibration.
>
> Our framework can generalize to other domains, because as long as a confidence score is available, we can conduct the four basic types of attacks we proposed, by applying a corresponding perturbation (e.g. word swap in text) that affects the confidence score but does not alter the label (e.g. by rejecting perturbations that flip the label). We think the interesting property of NLP in this task is its discrete property and the measurement of semantic similarity and semantic relatedness, which by themselves (as in many previous works) guarantee other stand-along papers. For example, due to the discrete nature of text tokens, formulating similarity metrics between an original and adversarial input that maintain the same meaning and the labels could be interesting, especially given that perturbations can range from simple character or word swaps to full sentence rephrases. Proposing ideal metrics to maintain semantics and the predicted labels (e.g., in the context of a generative model) would need specific studies. Our current paper, however, lays out the fundamental framework and four basic calibration attacking types that could be extended to other modalities.
>
> Reference:
>
> Ido Galil and Ran El-Yaniv. Disrupting deep uncertainty estimation without harming accuracy. In NeurIPS, 2021. URL https://openreview.net/forum?id=jGqcfSqOUR0.
>
> Aounon Kumar, Alexander Levine, Soheil Feizi, and Tom Goldstein. Certifying confidence via randomized smoothing. NeurIPS, 2020. URL https://par.nsf.gov/biblio/10207641.
>
> David Stutz, Matthias Hein, and Bernt Schiele. Confidence-Calibrated Adversarial Training: Generalizing to Unseen Attacks. ICML, 2020.

---

> ### Author Response · Authors · 2024-08-13
> **Response to Review nBsh (4/5)**
>
> **More Detailed Results**
>
> Below are more detailed results discussed in the above responses.
>
> Table 9: Attack effectiveness of UCA for the black box attacks across different model sizes on CIFAR-100.
>
> |||Avg \#q|Med. \#q|ECE|KS|Avg.  Conf.|
> |:----------------------------------|:---|:----------:|:----------:|:----------:|:----------:|:------------:|
> |ResNet18 (Accuracy: .854±.004)|||||||
> ||Pre|-|-|.037±.003|.016±.005|.870±.001|
> ||Post|60.5±1.5|35.3±1.5|.531±.005|.470±.006|.446±.007|
> |Resnet50 (Accuracy: .881±.002)|||||||
> ||Pre|-|-|.052±.006|.035±.006|.916±.006|
> ||Post|74.3±3.4|42.7±1.5|.540±.005|.479±.001|.465±.005|
> |ResNet152 (Accuracy: .883±.002)|||||||
> ||Pre|-|-|.051±.005|.046±.004|.929±.005|
> ||Post|90.4±5.5|48.0±1.0|.538±.007|.476±.004|.473±.003|
> |ViT-base (Accuracy: .935±.002)|||||||
> ||Pre|-|-|.064±.006|.054±.005|.882±.004|
> ||Post|118.5±2.4|62.0±3.1|.572±.007|.553±.004|.404±.003|
> |ViT-large (Accuracy: .936±0.001)|||||||
> ||Pre|-|-|.037±.009|.022±.012|.921±.021|
> ||Post|154.0±6.1|76.8±4.5|.536±.007|.514±.013|.444±.019|
> |Swin-Tiny (Accuracy: .883±.002)|||||||
> ||Pre|-|-|.040±.008|.031±.013|.914±.011|
> ||Post|109.1±6.5|54.5±2.2|.500±.012|.442±.003|.501±.006|
> |Swin-Small (Accuracy: .899±.008)|||||||
> ||Pre|-|-|.036±.007|.024±.008|.923±.001|
> ||Post|139.8±5.8|73.7±2.0|.507±.001|.456±.005|.496±.002|
> |Swin-Base (Accuracy: .939±.009)|||||||
> ||Pre|-|-|.034±.001|.012±.005|.940±.004|
> ||Post|191.8±18.2|104.7±14.5|.464±.012|.438±.010|.525±.015|
> |Swin-Large (Accuracy: .943±.005)|||||||
> ||Pre|-|-|.026±.006|.010±.003|.945±.005|
> ||Post|177.8±9.9|96±3.5|.455±.006|.430±.005|.534±.008|
>
>
> Table 10: Attack effectiveness of RCA for the black box attacks across different model sizes on CIFAR-100.
>
> |||Avg \#q|Med. \#q|ECE|KS|Avg. Conf.|
> |:----------------------------------:|:--:|:----------:|:-----------:|:----------:|:----------:|:------------:|
> |ResNet18 (Accuracy: .854±.004)|||||||
> ||Pre|-|-|.037±.003|.016±.005|.870±.001|
> ||Post|56.3±1.9|40.3±1.5|.479±.015|.420±.007|.495±.006|
> |Resnet50 (Accuracy: .881±.002)|||||||
> ||Pre|-|-|.052±.006|.035±.006|.916±.006|
> ||Post|68.9±4.6|42.7±1.2|.558±.011|.461±.003|.514±.003|
> |ResNet152 (Accuracy: .883±.002)|||||||
> ||Pre|-|-|.051±.005|.046±.004|.929±.005|
> ||Post|81.0±1.5|48.0±2.0|.512±.003|.451±.002|.496±.003|
> |ViT-base (Accuracy: .935±.002)|||||||
> ||Pre|-|-|.064±.006|.054±.005|.882±.004|
> ||Post|106.4±3.0|70.3±1.5|.549±.002|.505±.003|.471±.007|
> |ViT-large (Accuracy: .936±.001)|||||||
> ||Pre|-|-|.037±.009|.022±.012|.921±.021|
> ||Post|122.5±11.6|83.5±5.1|.502±.005|.479±.001|.479±.008|
> |Swin-Tiny (Accuracy: .883±.002)|||||||
> ||Pre|-|-|.040±.008|.031±.013|.914±.011|
> ||Post|101.4±.7|62.7±2.9|.474±.018|.417±.009|.526±.008|
> |Swin-Small (Accuracy: .899±.008)|||||||
> ||Pre|||.036±.007|.024±.008|.923±.001|
> ||Post|124.6±5.4|79.2±3.4|.483±.008|.432±.009|.519±.006|
> |Swin-Base (Accuracy: .939±.009)|||||||
> ||Pre|-|-|.034±.001|.012±.005|.940±.004|
> ||Post|161.6±25.5|116.6±21.3|.440±.011|.415±.009|.548±.010|
> |Swin-Large (Accuracy: .943±.005)|||||||
> ||Pre|-|-|.026±.006|.010±.003|.945±.005|
> ||Post|138.8±14.4|102.2±5.3|.433±.011|.409±.006|.557±.003|
>
> Table 11: Attack effectiveness of OCA for the black box attacks across different model sizes on CIFAR-100.
>
> |||Avg \#q|Med. \#q|ECE|KS|Avg. Conf.|
> |:----------------------------------:|:--:|:----------:|:-----------:|:----------:|:----------:|:------------:|
> |ResNet18 (Accuracy: .854±.004)|||||||
> ||Pre|-|-|.037±.003|.016±.005|.870±.001|
> ||Post|21.6±1.4|1.0±.0|.082±.003|.083±.003|.936±.003|
> |Resnet50 (Accuracy: .881±.002)|||||||
> ||Pre|-|-|.052±.006|.035±.006|.916±.006|
> ||Post|16.0±.8|1.0±.0|.124±.002|.124±.002|.996±.000|
> |ResNet152 (Accuracy: .883±.002)|||||||
> ||Pre|-|-|.051±.005|.046±.004|.929±.005|
> ||Post|10.3±2.6|1.0±.0|.077±.001|.077±.001|.961±.002|
> |ViT-base (Accuracy: .935±.002)|||||||
> ||Pre|-|-|.064±.006|.054±.005|.882±.004|
> ||Post|524.7±88.7|510.5±114.3|.043±.007|.043±.006|.974±.001|
> |ViT-large (Accuracy: .936±.001)|||||||
> ||Pre|-|-|.037±.009|.022±.012|.921±.021|
> ||Post|345.0±96.1|295.75±117.7|.039±.006|.039±.004|.962±.008|
> |Swin-Tiny (Accuracy: .883±.002)|||||||
> ||Pre|-|-|.040±.008|.031±.013|.914±.011|
> ||Post|27.3±9.8|1.0±.0|.074±.004|.075±.004|.958±.003|
> |Swin-Small (Accuracy: .899±.008)|||||||
> ||Pre|||.036±.007|.024±.008|.923±.001|
> ||Post|35.2±7.3|1.0±.0|.065±.005|.065±.005|.965±.002|
> |Swin-Base (Accuracy: .939±.009)|||||||
> ||Pre|-|-|.034±.001|.012±.005|.940±.004|
> ||Post|34.6±4.3|1.0±.0|.039±.004|.039±.005|.978±.005|
> |Swin-Large (Accuracy: .943±.005)|||||||
> ||Pre|-|-|.026±.006|.010±.003|.945±.005|
> ||Post|34.4±4.9|1.0±.0|.039±.002|.040±.001|.980±.004|

---

> ### Author Response · Authors · 2024-08-13
> **Response to Review nBsh (5/5)**
>
> Table 12: Attack effectiveness of MMA for the white box attacks on different model sizes on CIFAR-100.
>
> |||ECE|KS|Avg. Conf. |
> |:--------:|:--:|:-----------:|:------------:|:------------:|
> |ResNet18|Pre|.037±.003|.016±.005|.870±.001|
> ||Post|.158±.006|.144±.004|.726±.005|
> |Resnet50|Pre|.052±.006|.035±.006|.916±.006|
> ||Post|.187±.008|.161±.007|.746±.007|
> |ResNet152|Pre|.051±.005|.046±.004|.929±.005|
> ||Post|.157±.011|.137±.007|.772±.002|
> |ViT-base|Pre|.064±.006|.054±.005|.882±.004|
> ||Post|.260±.006|.262±.007|.675±.005|
> |ViT-large|Pre|.037±.009|.022±.012|.921±.021|
> ||Post|.279±.054|.277±.056|.662±.056|
> |Swin-Tiny|Pre|.040±.008|.031±.013|.914±.011|
> ||Post|.078±.005|.059±.015|.842±.018|
> |Swin-Small|Pre|.036±.007|.024±.008|.923±.001|
> ||Post|.117±.005|.101±.008|.817±.005|
> |Swin-Base|Pre|.034±.001|.012±.005|.940±.004|
> ||Post|.160±.035|.154±.032|.795±.024|
> |Swin-Large|Pre|.026±.006|.010±.003|.945±.005|
> ||Post|.135±.009|.133±.012|.827±.013|
>
>
> Table 13: Attack effectiveness of UCA for the white box attacks on different model sizes on CIFAR-100.
>
>
> |||ECE|KS|Avg. Conf.|
> |:--------:|:--:|:-----------:|:------------:|:-----------:|
> |ResNet18|Pre|.037±.003|.016±.005|.870±.001|
> ||Post|.232±.012|.176±.006|.740±.003|
> |Resnet50|Pre|.052±.006|.035±.006|.916±.006|
> ||Post|.213±.003|.175±.007|.747±.011|
> |ResNet152|Pre|.051±.005|.046±.004|.929±.005|
> ||Post|.210±.012|.158±.003|.781±.007|
> |ViT-base|Pre|.064±.006|.054±.005|.882±.004|
> ||Post|.277±.001|.274±.004|.671±.006|
> |ViT-large|Pre|.037±.009|.022±.012|.921±.021|
> ||Post|.299±.058|.285±.063|.668±.069|
> |Swin-Tiny|Pre|.040±.008|.031±.013|.914±.011|
> ||Post|.142±.023|.088±.021|.854±.017|
> |Swin-Small|Pre|.036±.007|.024±.008|.923±.001|
> ||Post|.179±.004|.128±.008|.829±.005|
> |Swin-Base|Pre|.034±.001|.012±.005|.940±.004|
> ||Post|.217±.019|.175±.023|.812±.020|
> |Swin-Large|Pre|.026±.006|.010±.003|.945±.005|
> ||Post|.191±.015|.158±.01|.816±.017|
>
>
>
> Table 14: Attack effectiveness of RCA for the white box attacks on different model sizes on CIFAR-100.
>
> |||ECE|KS|Avg. Conf.|
> |:--------:|:--:|:-----------:|:------------:|:------------:|
> |ResNet18|Pre|.037±.003|.016±.005|.870±.001|
> ||Post|.179±.016|.138±.006|.762±.003|
> |Resnet50|Pre|.052±.006|.035±.006|.916±.006|
> ||Post|.187±.016|.156±.013|.759±.015|
> |ResNet152|Pre|.051±.005|.046±.004|.929±.005|
> ||Post|.188±.004|.144±.008|.789±.016|
> |ViT-base|Pre|.064±.006|.054±.005|.882±.004|
> ||Post|.236±.013|.239±.015|.699±.013|
> |ViT-large|Pre|.037±.009|.022±.012|.921±.021|
> ||Post|.265±.047|.259±.051|.686±.057|
> |Swin-Tiny|Pre|.040±.008|.031±.013|.914±.011|
> ||Post|.103±.0140|.067±.018|.855±.018|
> |Swin-Small|Pre|.036±.007|.024±.008|.923±.001|
> ||Post|.148±.014|.112±.015|.832±.010|
> |Swin-Base|Pre|.034±.001|.012±.005|.940±.004|
> ||Post|.175±.022|.152±.019|.816±.013|
> |Swin-Large|Pre|.026±.006|.010±.003|.945±.005|
> ||Post|.159±.014|.136±.014|.842±.020|
>
> Table 15: Attack effectiveness of OCA for for the white box attacks on different model sizes on CIFAR-100.
>
> |||ECE|KS|Avg. Conf.|
> |:--------:|:--:|:-----------:|:-----------:|:------------:|
> |ResNet18|Pre|.037±.003|.016±.005|.870±.001|
> ||Post|.079±.006|.079±.006|.933±.002|
> |Resnet50|Pre|.052±.006|.035±.006|.916±.006|
> ||Post|.072±.003|.070±.001|.951±.002|
> |ResNet152|Pre|.051±.005|.046±.004|.929±.005|
> ||Post|.075±.001|.075±.002|.959±.001|
> |ViT-base|Pre|.064±.006|.054±.005|.882±.004|
> ||Post|.045±.003|.017±.002|.928±.002|
> |ViT-large|Pre|.037±.009|.022±.012|.921±.021|
> ||Post|.039±.011|.021±.008|.928±.019|
> |Swin-Tiny|Pre|.040±.008|.031±.013|.914±.011|
> ||Post|.070±.004|.066±.005|.949±.003|
> |Swin-Small|Pre|.036±.007|.024±.008|.923±.001|
> ||Post|.057±.008|.056±.008|.955±.001|
> |Swin-Base|Pre|.034±.001|.012±.005|.940±.004|
> ||Post|.037±.005|.036±.003|.974±.006|
> |Swin-Large|Pre|.026±.006|.010±.003|.945±.005|
> ||Post|.038±.003|.037±.002|.978±.004|

---

### Review · Reviewer_RYXa · 2024-08-14

**Summary Of Contributions:**

The authors propose a family of black-box and white-box adversarial attacks on the calibration of a neural network model. Specifically, they consider crafting adversarial examples causing overconfidence, underconfidence, random confidence or cause maximum miscalibration across test examples. They also consider adversarially training models against such attacks as well as a post-calibration defense based on temperature scaling.

**Audience:**

Yes

**Broader Impact Concerns:**

No concerns.

**Claims And Evidence:**

No

**Requested Changes:**

See weaknesses.

**Strengths And Weaknesses:**

Strengths:
- Generally interesting question posed by the paper: how can we attack (and defend) against attackers that care about some sort of uncertainty/confidence of the model
- Clean writing and presentation
- Fairly thorough experimental setup, including several baselines, adversarial example detection, white- and black-box attack, etc.

Weaknesses:
- Generally, I am not convinced by the set up of the calibration attacks studies in this paper. Mainly, I believe that uncertainty estimates such as confidence are only useful in a decision-making context. While ECE is a useful measure for calibration, having a low ECE is only relevant in practice if the confidence is actually used for something. This could be abstention, deferral to another system or human, OOD detection or similar detection settings, human rater studies, etc. Without a downstream use-case (i.e., decision problem), the confidence does not matter much. In classification, without a downstream use of the confidence, the actual classification task does not care about the confidence being calibrated or not. I feel this paper misses the point to some extent, focusing on confidence attacks without any downstream task, using ECE as a proxy. Instead, I would expect experiments on downstream tasks or potentially alternative approaches to calibration and measuring calibration (e.g., from the conformal prediction literature [1] or a Bayesian perspective).

[1] https://arxiv.org/abs/2107.07511

- The attacks themselves do not introduce any new tools or improve over existing besides changing the objectives. It is well known that many attacks, especially PGD, are widely applicable. For example, [2] already had an attack targeting confidence (but still expecting mis-classification) and highlighting some of the optimization problems that come with a different objective. Btw, [2] also has a similar adversarial training routine that would be an appropriate baseline.

[2] https://arxiv.org/abs/1910.06259

- Regarding the CS defense I am wondering why this procedure does not impact the calibration on clean examples? Table 4 shows that pre attack ECE is low with CS, but if we increase the confidence of low confidence examples, ECE should increase more significantly. In any case, it severely changes the calibration property on clean examples which should be a no-go in many practical applications where we care about calibration.
- Where the models used for attacking and as defenses calibrated in any way before the experiments? I couldn’t find that information in appendix C but would expect that all models where calibrated using e.g. vector or temperature scaling prior to experiments.
- Tables 1 and 5 suggest that black-box attacks are actually more successful in worsening the ECE. Did I understand that correctly? If so, why? White-box attacks should be significantly more powerful and the difference in ECE is significant. Is this an optimization problem?
- The proposed AT variant does not improve over baseline AT. Only the accuracy-ECE trade-off is better – the proposed method has worse robustness in terms of ECE but slightly better accuracy. This trade-off is not explored.
- Detecting white-box examples is known to be flawed. [2] and other works repeatedly showed that the considered methods are not robust in the white-box setting when considering appropriate attacks. At this point, the experiments merely show that confidence attacks are easier to detect; or they might show that the confidence attacks are not tuned enough to be successful. In any case, these experiments do not add a lot of insights.

Conclusion:
While I like the general direction, I think the paper focused on the wrong experimental setup by studying impact on ECE of such attacks rather than actually relevant downstream tasks (e.g., abstention, detection, deferral, human-AI teams, other decision making setups). Moreover, the experiments have some flaws and do not provide many insights beyond general intution about adversarial attacks.

---

> ### Author Response · Authors · 2024-08-25
> **Response to Review RYXa (1/4)**
>
> We thank the reviewer for the comments.
>
> **Reviewer’s Comment 1**:  “I believe that uncertainty estimates such as confidence are only useful in a decision-making context. While ECE is a useful measure for calibration, having a low ECE is only relevant in practice if the confidence is actually used … … ”
>
> **Response**:
>
> Confidence could be critical in affecting decision-making outcomes themself but also the trust of humans in general. We respectfully argue that this does not mean the evaluation must be performed in an extrinsic downstream task. It is common practice that in many tasks in image and natural language processing, intrinsic evaluations are performed, where the metrics themselves do not directly reflect downstream utility. One advantage of doing so is isolating a task from others to avoid complications and, sometimes, avoid less conclusive claims (particularly when the downstream tasks are very diverse, which is our case here).
>
> Specifically for the problem we study in this paper, the importance of having low miscalibration in downstream applications has been well established in previous work, and the adverse impact of large calibration errors on applications is clear, when the labels in classification are not modified. Below we discuss the related previous work.
>
> Previous studies (Penso et al, 2024; Jiang et al, 2012; Kompa et al, 2021) have shown the effect of miscalibration on down-stream authority mechanisms, even when the labels in classification are not changed. For example, Penso et al. (2024) discuss how medical imaging, if skipping expert review for the confident but incorrect model predictions, can have disastrous consequences. Overconfidence attacks as well as their combinations with underconfidence attacks can inherently change the reviewing lists suggested to the experts, without changing the (incorrect) class labels, and then harm the decision process. As another example, Kompa et al. (2021) advocate for medical AI models being able to abstain from providing a diagnosis when there is a significant uncertainty for a patient, and to defer to additional human expertise to make a more informed diagnosis. This relies on a model that can give well-calibrated estimates. Skewing confidences can directly change the behavior of such systems.
>
> Moreover, prior works have examined the role of confidence in affecting the level of trust humans have in a system, which influences applications where humans and AI decision-making occurs symbiotically (Zhang et al, 2020; Rechkemmer et al, 2022). For example, Zhang et al. (2020) demonstrate when a model produces high confidence scores, users are more prone to rely on the AI for decision-making. Antifakos et al. (2005), through a study on a context-aware setup, show how providing explicit confidence changes user trust in a system. Skewing confidence using calibration attacks inherently influences the process and jeopardizes human trust in such systems.
>
>
> As more specific examples in the healthcare domain, chest abnormality detection (Dyer, 2021) tag radiographs with high-confidence normality to help radiologists make decisions. In child maltreatment hotlines, screening models (De-Arteaga, 2020) are deployed where confidence is provided to suggest the likelihood that a child on the referral will experience adverse welfare-related outcomes. Call workers use confidence, along with other information, to make their decisions. The study discusses how the tool could be “glitching” out by providing incorrect confidence, compromising operability. Calibration attacks can easily jeopardize the operability. In the legal domain, LegalLens (Bernsohn, 2024) detects legal violations. Entities with high confidence are further processed with downstream models. In enterprise security, anomalous behaviour detectors (Shashanka, 2016) use confidence to send alerts about likely malicious activity.
>
> **Based on the reviewer’s comments, we will add the above discussion into our revision.** Thank you.
>
> The feedback from the reviewer also encourages us to ponder **whether the current study has provided adequate contributions on its own**. We respectfully argue that is the case, although additional extrinsic evaluations, if being properly set up with well-defined target hypotheses, could invite additional studies.
>
> First, the contributions of the current paper, which provide the first comprehensive study on calibration attacks, could jeopardize any downstream tasks that rely on the victim models’ confidences, including those discussed in the above literature, where the impact of skewed confidence has been well understood. In addition to attacks, the paper also provides a comprehensive study on a wide range of adversarial defence and recalibration methods to mitigate the harm.

---

> > ### Author Response · Authors · 2024-08-25
> > **Response to Review RYXa (2/4)**
> >
> > Second, we would respectfully argue that the intrinsic evaluations we perform are adequate to support the claims and contributions of this paper. The paper will be over 30 pages including appendices after revision and we believe it has made stand-along contributions on its own due to the reasons discussed above. We also believe that skewing confidence and defending against it are a critical topic.
> >
> > Third, in this paper we focused on intrinsic evaluation. We think that using the most typical metrics (e.g., ECE) help make our work comparable to existing and future works.
> >
> > ----------------------
> > References:
> >
> > M. Andriushchenko, et al. Square Attack: a query-efficient black-box adversarial attack. ECCV 2020.
> >
> > P. Chen, et al. ZOO: Zeroth Order Optimization Based Black-Box Attacks to Deep Neural Networks without Training Substitute Models. AISEC. 2017.
> >
> > C. Guo, et al. On Calibration of Modern Neural Networks. ICML, 2017
> >
> > X. Jiang, et al. Calibrating predictive model estimates to support personalized medicine. JAMIA, 19:263–74, 2012.
> >
> > B. Kompa, et al. Second opinion needed: communicating uncertainty in medical machine learning. Digital Medicine, 4, 2021.
> >
> > C. Penso, et al. Confidence calibration of a medical imaging classification system that is robust to label noise. IEEE Tran. Med. Imag., 43:2050–60, 2024.
> >
> > A. Kumar, et al. Trainable Calibration Measures for Neural Networks from Kernel Mean Embeddings. ICML, 2018.
> >
> > A. Madry, et al. Towards Deep Learning Models Resistant to Adversarial Attacks. ICLR, 2018.
> >
> > M. Minderer, et al. Revisiting the Calibration of Modern Neural Networks. ArXiv, abs/2106.07998, 2021.
> >
> > A. Rechkemmer and M. Yin. When confidence meets accuracy: Exploring the effects of multiple performance indicators on trust in machine learning models. CHI, 2022.
> >
> > C. Xie, et al. Improving transferability of adversarial examples with input diversity. CVPR, 2019

---

> ### Author Response · Authors · 2024-08-25
> **Response to Review RYXa (3/4)**
>
> **Reviewer’s Comment 2**: “The attacks themselves do not introduce any new tools or improve over existing besides changing the objectives. It is well known that many attacks, especially PGD, are widely applicable. … …”
>
> **Response**:
>
> We respectfully argue that this undercuts the extensive study that we have performed, for example, the introduction of the MCA and RCA attacks, which aim to create more complex patterns of miscalibration (without changing the accuracy). We provided a comprehensive study for calibration attacks targeting confidence, and for optimizing miscalibration through said attacks, which of course, as the reviewer commented on, is based on changing the objective. But we believe the objective is very important. We did not claim that we are creating a brand-new form of adversarial attack that is better than some other, so we should not be judged for not doing this. Instead, we highlight how easy it is to adapt existing attacks to perform confidence attacks that can create very high levels of miscalibration without changing the accuracy and being more difficult to detect, as well as how a series of calibration methods and adversarial defenses fare. In addition, the work of (Stutz et al., 2020) is quite different from our own, and does not explore comprehensive attacks targeting calibration as we have. Their methodology focuses on a modified adversarial training objective which focuses on biasing adversarial samples to be low-confidence predictions, an orthogonal contribution compared to our work.
>
> **Reviewer’s Comment 3**: "Regarding the CS defense I am wondering why this procedure does not impact the calibration on clean examples? Table 4 shows that pre attack ECE is low with CS, but if we increase the confidence of low confidence examples, ECE should increase more significantly. In any case, it severely changes the calibration property on clean examples which should be a no-go in many practical applications where we care about calibration."
>
> **Response**:
>
> As we show in the paper, the effect on pre-attack ECE is low in practice (presumably because very low-confidence samples are extremely rare for these models, which possess very high confidence and accuracy by default). There are certainly limitations to using CS in practical applications, which is why we do not universally recommend it as a solution to calibration attacks. However, the method provides insights into what could be effective against such attacks that are not just related to general adversarial robustness, and thus we believe it is of interest.

---

> ### Author Response · Authors · 2024-08-25
> **Response to Review RYXa (4/4)**
>
> **Reviewer’s Comment 4**:  "Where the models used for attacking and as defenses calibrated in any way before the experiments? I couldn’t find that information in appendix C but would expect that all models where calibrated using e.g. vector or temperature scaling prior to experiments."
>
> **Response**:
>
> Prior to attacks, all the models did not undergo any calibration or scaling methods as we wished to test the influence of these kinds of methods against the attacks. The baselines where we test calibration methods do incorporate the corresponding methods (e.g. temperature scaling).
>
>
> **Reviewer’s Comment 5**: "Tables 1 and 5 suggest that black-box attacks are actually more successful in worsening the ECE. Did I understand that correctly? If so, why? White-box attacks should be significantly more powerful and the difference in ECE is significant. Is this an optimization problem?"
>
>
> **Response**:
>
>
> As we explain in the appendix when describing the white-box results, this is because white-box attacks are often too effective at inducing label flips. Therefore, only perturbations that maintain the original label are accepted, as the attack is optimizing for confidence drops, which ends up being less efficient than the original algorithm. In contrast, SA is very effective at gradually lowering confidence without flipping the label, allowing it to generally perform better. It is also important to note that the number of attack iterations between the two setups is significantly different, e.g., in many cases 1000 vs. 10, which also influences the effectiveness of the white-box variant.
>
>
> **Reviewer’s Comment 6**:  "The proposed AT variant does not improve over baseline AT. Only the accuracy-ECE trade-off is better – the proposed method has worse robustness in terms of ECE but slightly better accuracy. This trade-off is not explored."
>
>
> **Response**:
>
>
> The method is not intended to be an improvement over baseline AT; instead, we propose it to test the hypothesis of whether using calibration-attacked samples leads to an improvement in resistance compared to regularly adversarially attacked data in the training objective. As we can see from the results, AT is stronger since it focuses on general adversarial robustness. CAAT does not train on samples where perturbations lead to label changes, so the inductive bias towards resisting the attacks is lessened, resulting in better accuracy at the tradeoff of lesser adversarial robustness. This shows that exposure to calibration-attacked data during adversarial training does not lead to improved resistance to calibration attacks; instead, prioritizing general adversarial robustness in this case seems to be more effective.
>
>
> **Reviewer’s Comment 7**:  "Detecting white-box examples is known to be flawed. [2] and other works repeatedly showed that the considered methods are not robust in the white-box setting when considering appropriate attacks. At this point, the experiments merely show that confidence attacks are easier to detect; or they might show that the confidence attacks are not tuned enough to be successful. In any case, these experiments do not add a lot of insights."
>
>
> **Response**:
>
>
> We have tested an array of very widely used and well cited detection methods to compare how detectable calibration attacks are compared to regular adversarial attack variants, so although these methods are not perfect (no methods are), they give a good idea as to how the detectability is across these two setups, and many papers use them as baselines to test the detectability of methods. In addition, we do not only detect white-box examples but also black-box ones, and our results remain consistent across both and across all tested methods. The results do not show that confidence attacks are easier to detect; on the contrary, we show they are harder to detect, even though they are highly effective at inducing miscalibration as we also demonstrate. This adds insights as it proves that they are more challenging to detect and thus more stealthy.

---

### Decision · Action_Editor_UWHP · 2024-10-25

**Recommendation:** Accept with minor revision

**Comment:**

Overall, reviewers were supportive of acceptance, subject to the two main suggestions above: more discussion of the significance of the problem, demonstrating the downstream impact of the attacks. Both would undoubtedly strengthen the paper; we believe though that even without the latter, the paper could be of sufficient interest as it is. Nonetheless, it is important that the authors add a discussion on this point.

**Requested changes**

- add discussion of the significance of the problem, in Section 3.4 or elsewhere. This can be a summary of the response to Reviewer 9Y8b.

- add discussion on the focus of ECE versus downstream metrics.

- add experimental results on the transferrability of the attacks across models, per the response to Reviewer 9Y8b.

- add experimental results on Swin Transformers, per the response to Reviewer nBsh.

- typo: page 3, "nubmer"

- formatting: x-axis for Fig 1 and legend for Fig 2 are quite small.

- formatting: in Equation 3, e.g., suggest to use \rm font for the subscript.

**Audience:**

Reviewers were in agreement that the new form of adversarial attack studied here could be of interest to the community: given the importance of uncertainty estimation in real-world decision making, it is of broad interest to understand how vulnerable these estimates are to adversarial manipulation.

Reviewers did however suggest that the authors try to motivate more clearly the significance of this problem. This is related to the above comment on the downstream impact of such attacks. Such discussion could help broaden the visibility of this work.

**Claims And Evidence:**

The main claim of the paper is that it is possible to perform an adversarial attack targetting the _probabilistic calibration_ of a classifier (e.g., as measured by the expected calibration error). The paper explores four different types of attack in white- and black-box form, and studies the viability of existing defense mechanisms. The conclusion of these analyses is that existing mechanisms may not be sufficient to guard against these attacks.

Reviewers were in agreement that the paper offers an interesting new type of adversarial attack. A qualifying comment is that reviewers were keen on seeing the actual influence of these attacks on downstream decisions, rather than just on the calibration error. This would make a more compelling argument for the impact of such attacks. At the same time, this could conceivably be explored in future work. The authors also argued that their analysis is akin to studying perplexity in natural language modelling, as opposed to downstream metrics, which appears a reasonable argument.